# Snow Water Equivalent from airborne Ku-band data: The Trail Valley Creek 2018/19 Snow Experiment

Benoit Montpetit<sup>1</sup>, Julien Meloche<sup>1</sup>, Vincent Vionnet<sup>2</sup>, Chris Derksen<sup>1</sup>, Georgina Woolley<sup>3</sup>, Nicolas R. Leroux<sup>2</sup>, Paul Siqueira<sup>4</sup>, J. Max Adam<sup>4</sup>, and Mike Brady<sup>1</sup>

**Correspondence:** Benoit Montpetit (benoit.montpetit@ec.gc.ca)

Abstract. Snow is an important freshwater resource that impacts the health and well-being of communities, the economy, and sustains ecosystems of the cryosphere. This is why there is a need for a spaceborne Earth observation mission to monitor global snow conditions. Environment and Climate Change Canada, in partnership with the Canadian Space Agency, is developing a new Ku-band synthetic aperture radar mission to retrieve snow water equivalent (SWE) at a nominal resolution of 500 m, and weekly coverage of the cryosphere. Here, we present the concept of the SWE retrieval algorithm for this proposed satellite mission. It is shown that by combining a priori knowledge of snow conditions from a land surface model, like the Canadian Soil Vegetation Snow version 2 model (SVS-2), in a Markov Chain Monte Carlo (MCMC) Bayesian model coupled with the Snow Microwave Radiative Transfer model (SMRT), we can retrieve SWE with an RMSE of 15.8 mm (16.4 %) and a MCMC-retrieved SWE uncertainty of 23.4 mm (25.2 %). To achieve this accuracy, a larger uncertainty in the a priori grain size estimation is required, since this variable is known to be underestimated within SVS-2 and has a considerable impact on the microwave scattering properties of snow. It is also shown that adding four observations from different incidence angles improves the accuracy of the SWE retrieval because these observations are sensitive to different scattering mechanisms of the snowpack. These results validate the mission concept of the proposed Canadian satellite mission.

#### 1 Introduction

Yearly, snow can cover more than 50 % of the terrestrial northern hemisphere (Robinson et al., 2012) and is an important fresh water resource that impacts the health and well-being of communities, the economy, and sustains ecosystems (Meredith et al., 2019). Snow extent and mass trends are forecasted to keep decreasing at a rate up to -50 x 10<sup>6</sup> km<sup>2</sup> year<sup>-1</sup> and -5 Gt year<sup>-1</sup>, respectively (Mudryk et al., 2020). Yet, it is still the only component of the water cycle that, currently, does not have a dedicated Earth Observation (EO) mission (Derksen et al., 2019). Monitoring snow water equivalent (SWE), i.e. the amount of water stored in solid or liquid form in the snowpack, at high spatio-temporal resolution is critical for climate services, water resource management, and environment prediction (Garnaud et al., 2019; Kim et al., 2021; Cho et al., 2023). Following the work done for the European Space Agency (ESA) Earth Explorer 7 Cold Regions Hydrology High-resolution

<sup>&</sup>lt;sup>1</sup>Climate Research Division, Environment and Climate Change Canada, Ontario, Canada

<sup>&</sup>lt;sup>2</sup>Meteorological Research Division, Environment and Climate Change Canada, Quebec, Canada

<sup>&</sup>lt;sup>3</sup>Northumbria University, Newcastle upon Tyne, United Kingdom

<sup>&</sup>lt;sup>4</sup>College of Engineering, University of Massachusetts Amherst, MA, United States

Observatory (CoReH2O) mission (Rott et al., 2010), and recent work in the field of microwave snow remote sensing (Tsang et al., 2022), Environment and Climate Change Canada (ECCC), in partnership with the Canadian Space Agency (CSA), are developing a synthetic aperture radar (SAR) satellite mission that aims at imaging the Northern Hemisphere at a nominal resolution of 500 m on a weekly basis, currently named, the Terrestrial Snow Mass Mission (TSMM) (Derksen et al., 2019).

The international snow community has made considerable progress in the recent decade in demonstrating that Ku-Band radar measurements provide the best option for future satellite missions to monitor snow as a water resource at sub-kilometre spatial resolution, due to its sensitivity to SWE via its volume scattering in dry snow and its sensitivity to its phase (wet/dry) (Tsang et al., 2022). Even though passive microwave measurements show the same sensitivity to SWE and snow phase, the technology does not currently provide sub-kilometre measurements (Galeazzi et al., 2023). It is also known that, due to the sensitivity of the Ku-band radar backscatter ( $\sigma^0$ ) to the snow microstructure (King et al., 2018; Picard et al., 202b; Montpetit et al., 2024), retrieving SWE from a single microwave measurement can prove challenging (Lemmetyinen et al., 2018; Pan et al., 2024). This is why TSMM presents a dual Ku-Band frequency (13.25 and 17.5 GHz), dual polarization (VV/VH) concept to constrain a retrieval algorithm with more measurements, i.e. the higher Ku frequency being more sensitive to snow microstructure than the lower frequency and the cross-polarization signal being more sensitive to interactions within the snow volume than the co-polarization (Ulaby and Ravaioli, 2020). The main objective of TSMM is to retrieve SWE from these satellite observations with a seasonal root-mean-square error (RMSE) of 25 % in alpine regions and 30 mm elsewhere (Derksen et al., 2019). These observations will then be ingested into the Canadian Land Data Assimilation Scheme (CaLDAS) (Carrera et al., 2015; Garnaud et al., 2021) in order to improve ECCC's numerical weather/climate prediction services. Assimilating TSMM retrievals will also help improve surface modelling like the Canadian Soil Vegetation Snow (SVS) (Leonardini et al., 2021) model and other hydrological systems such as the Canadian Hydrological Model (CHM) (Marsh et al., 2020). This study aims at developing the workflow that will be used to derive SWE from the dual-frequency SAR measurements and also provide stratified snow information that will be crucial to improve hydrological and land surface modelling via data assimilation across all the various landscapes found in Canada.

Many studies have developed Bayesian methods to retrieve SWE from SAR (Rott et al., 2012; Singh et al., 2024; Pan et al., 2024). It is key for these methods to correctly specify SWE uncertainty, where it was achieved by specifying layer and density uncertainties. Rott et al. (2012) used a constrained minimization approach where SWE and effective snow grain radius was optimized iteratively to match forward modelled and measured  $\sigma^0$ . This method was intended to be applied to X-band and Ku-band  $\sigma^0$  measurements for the CoReH2O mission. Singh et al. (2024) used a Bayesian inference model that seeks to estimate the joint probability of backscatter measurements and snow properties. Prior distributions of snow parameters were necessary for this approach and were obtained from a multilayered snow hydrological model driven by numerical weather prediction (NWP) forecasts. This method was also applied to X- and Ku-band SAR data and showed great success rate to retrieve SWE over Grand Mesa, Colorado, USA. Pan et al. (2024) modified the Bayesian-based Algorithm for SWE Estimation (Pan et al., 2017) to apply it to active microwave measurements. This methods relies on the Markov Chain Monte Carlo (MCMC) method to optimize multiple snow properties simultaneously to minimize a cost function between the measured and forward modelled  $\sigma^0$ . They showed that an RMSE below 30 mm of SWE could be achieved when applied to X- and Ku-band data.

This study uses a Bayesian approach on data acquired during the 2018/19 Trail Valley Creek (TVC) experiment, where only single frequency Ku-band VV polarization data was acquired from an airborne platform (Montpetit et al., 2024). Since it has been largely documented that using a multi-layered snowpack approach considerably improves SWE retrievals compared to single layer snowpack (Pan et al., 2024; Durand et al., 2024; Singh et al., 2024; Lemmetyinen et al., 2018), this study only focuses on a multi-layered approach. We focus on the two dominant snow layers observed in an Arctic snowpack, i.e. a dense wind compacted rounded grains (R) snow layer at the surface with a coarse depth hoar (DH) layer at the bottom (Montpetit et al., 2024; Rutter et al., 2019; Derksen et al., 2009). The retrieval algorithm developed for this study was inspired by previous work using the MCMC method (Pan et al., 2024; Picard et al., 2022a; Pan et al., 2017). Section 3 details how the approach used in this study differs from previous work. In the context of an EO algorithm development, emphasis will be given on the need to include quality spatio-temporal information. Methods to improve computation efficiency, without compromising retrieval accuracy will also be presented.

Section 2 briefly describes the 2018/19 TVC experiment. For a more detailed explanation, please refer to Montpetit et al. (2024). Section 3.5 details the SWE retrieval architecture as well as the processing applied to field measurements in order to properly compare the outputs of the retrieved MCMC snow properties with surveyed properties in the field. Section 4.1 compares the Canadian land surface model Canadian Soil Vegetation Snow version 2 (SVS-2) outputs (Vionnet et al., 2025; Woolley et al., 2024) to field measurements, while section 4.2 shows the results to validate the MCMC approach. Sections 4.3 and 4.4 show the comparisons of the MCMC retrieved SWE and vertical snow properties to the surveyed properties. The efficiency of the MCMC method to retrieve SWE is assessed in section 5. Considerations in order to estimate both SWE and snow properties that are representative of actual snow conditions on the ground and the usage of SVS-2 and its future improvements to be implemented are also discussed in section 5.

#### 2 The Trail Valley Creek 2018/19 Snow Radar Experiment

The TVC 2018/19 experiment was designed by ECCC to advance science readiness activities for TSMM. The TVC watershed, near Inuvik, Northwest Territories, Canada, was selected since many snow and hydrological research activities are conducted there every year (e.g., Shi et al., 2015; Wilcox et al., 2022). Including the airborne SAR campaign for this study (Siqueira et al., 2021), there has been other similar campaigns over TVC like the SnowSAR campaign of 2012/13 (King et al., 2018) and more recently, in April 2024, the Cryospheric SAR (CryoSAR) instrument (Kelly et al., 2024) onboard the Alfred Wagner Institute (AWI) Polar 5 (Haas et al., 2024) was flown with a dual L- and Ku-band SAR. Other work at TVC focused on improving land surface modelling of Arctic environments (Woolley et al., 2024) using the Ensemble System Crocus (ESCROC) model (Lafaysse et al., 2017), which was implemented in the SVS-2 land surface model (Vionnet et al., 2025, 2022; Garnaud et al., 2019).

In a first step, Montpetit et al. (2024) has shown that the Ku-band radar instrument developed by the University of Massachusetts (UMASS) team (Siqueira et al., 2021) is sensitive to snow physical properties and that the Snow Radiative Transfer Model (SMRT) could be used to properly model the  $\sigma^0$  from surveyed snow properties. In this study, we will show that from

the same airborne  $\sigma^0$  measurements we can retrieve SWE using independent modelled data (SVS-2) as priors in the retrieval algorithm.

Figure 1 shows a map of the study area and the surveyed sites. The radar image in Figure 1 (left) consists in a mosaic of two different airborne passes, flown in the same direction, acquired by the UMASS radar system (Section 2.1), where the near range acquisitions (higher backscatter) of the first pass, done at steeper incidence angles, meets the far range acquisitions (lower backscatter) of the second pass, made at shallower incidence angles. A DEM (center) from the ArcticDEM (Porter et al., 2023) and the vegetation classification (right, Grünberg and Boike, 2019) is shown for context with the radar imagery. For a detailed description of the different dataset measured during this TVC experiment, please refer to Montpetit et al. (2024). Elements relevant to this study will be presented here.

**Figure 1.** Sites sampled during the January campaign of the TVC 2018/19 experiment. Squares correspond to a 100 m x 100 m around the central surveyed snowpit (see Section 2.2). Background images are two overlapped UMASS Ku-Band radar images corresponding to two different flight passes acquired November 14, 2018 (left, Siqueira et al., 2021), the 2 m ArcticDEM (center, Porter et al., 2023), and the vegetation classification (right, Grünberg and Boike, 2019).

### 2.1 Airborne SAR measurements

100

105

For this TVC experiment, the UMASS Ku-band SAR instrument was mounted on a Cessna-208. It flew at a nominal altitude of 1000 m, and measured  $\sigma^0$  at 13.285 GHz in VV polarization over a 2 km swath, with a 2 m ground-range resolution and an incidence angle range of ~20-70°. Flight lines were planned to maximize repeat coverage of the SikSik sub-basin within the TVC watershed. This enabled swath overlap between flight passes and measurements of the same targets in different viewing geometries. A total of 16 flight lines were planned, measuring selected targets within the area of interest (AOI) in four different look-directions. To compare measured  $\sigma^0$  to surveyed snow information, a 100 m x 100 m area was clipped around the surveyed site, was filtered to reduce noise and artifacts, and averaged.

Due to challenging flight conditions in November 2018 and challenging snow conditions in March 2019, only the January 2019  $\sigma^0$  measurements are used in this study to validate the SWE retrieval algorithm in dry snow conditions.

#### 110 2.2 Ground based snow and soil measurements

Within the AOI, six static sites were identified, in order to monitor the underlying ground conditions of the SikSik sub-basin throughout the winter, and also monitor the evolution of snow conditions over contrasting land covers, representative of TVC (Figure 1). Four HydroProbe soil sensors were installed horizontally in a soil pit at each of these static sites, where soil temperature, moisture and permittivity were measured continuously during the campaign. This data enabled the retrieval of microwave background soil properties from TerraSAR-X and RADARSAT-2 satellite SAR measurements (Montpetit et al., 2024).

Figure 2. Ground based snow measurements sampling scheme taken from Montpetit et al. (2024).

120

A total of 20 surveyed sites, including the static sites (Figure 1), are used in this study to validate the SWE retrieval algorithm. At the center of each of these sites, a snowpit was excavated, and a full snow profile was surveyed to use as reference snow measurements for a given site using the standard methods (Fierz et al., 2009). For each snowpit, snow temperature, density, Specific Surface Area (SSA) were measured at the pit wall. Density was measured using a Taylor-LaChapelle style cutter and a shielded digital scale. SSA was measured using the A2 Photonics IceCube instrument (Domine et al., 2007; Gallet et al.,

2009). Behind the pit wall, three Snow Micro Penetrometer (SMP) profiles were acquired (Proksch et al., 2015) in order to calibrate the force measurements to the reference density and SSA measurements. To get a representative distribution of snow microstructure at the airborne spatial scale, north-south and east-west transects were surveyed with the SMP (a total of 16 more profiles) covering an area of  $\sim$ 100 m x 100 m. Snow depth measurements were surveyed every meter along these transects with a MagnaProbe (Sturm and Holmgren, 2018) ( $\sim$ 290 measurements per site). Figure 2 shows the schematic of a typical sampling done for a given site. All profiles (snowpits, SMP and MagnaProbe profiles) per site are then used to generate a statistical representation of snow conditions within the radar footprint, with a measured snow uncertainty represented by the spatial variability within the footprint. Spatial variability consists in the largest uncertainty within the footprint compared to snow parameter measurement uncertainty, the latter can thus be neglected.

## 2.3 Soil Vegetation Snow version 2 model outputs

The SVS-2 model outputs used in this study are a subset of the dataset generated by Woolley et al. (2024). This dataset was generated for the period of September 1991 to September 2023, but only the period of January 12 to 15, 2019 was used, which corresponds to the three day window where the UMASS airborne SAR measurements were acquired during the January intensive campaign of this TVC experiment. These outputs were generated from point-scale simulations located at the main meteorological site of TVC (SM site in Figure 1), where most of the meteorological forcing data was acquired. and complemented by neighbouring stations when data was not available. The multi-layered snow information comes from the ESCROC model (Lafaysse et al., 2017). The one-dimensional, vertical snow profile outputs of Crocus, consist in mass, density, temperature, liquid water content, age and snow microstructure properties (optical diameter, sphericity) for each layer. These outputs can then be translated into thickness, density and SSA for each layer. The maximum number of layers was set to 20 for this dataset, in order to get detailed stratigraphic information. A total of 120 different simulations were conducted with different combinations of wind and surface vegetation effects, and thermal conductivity parameterizations. These ensembles were used to generate the priors for the MCMC retrieval algorithm (see Section 3.5.1). Test were also conducted in this study with the 30 ensemble members that had the best continuous ranked probability score (CRPS, see Woolley et al., 2024). Both versions used in the study of Woolley et al. (2024) are tested in this study, where an Arctic version of SVS-2 was developed to improve the overall snow properties and stratigraphy of Arctic snowpacks. For a complete description of the dataset, please refer to Woolley et al. (2024).

# 3 Methods

150

125

130

135

In this section, the SWE retrieval workflow will be presented as well as the methodology to compare the retrieved SWE data with measured data from the TVC 2018/19 experiment. In order to improve computation efficiency, the methodology to reduce the surveyed snowpack stratigraphy to two layers, will be introduced. A different approach, more automated (Meloche et al., 2025), which is applied to the SVS-2 outputs will also be described. Finally, the Bayesian MCMC methodology will be described in details in order to retrieved the SWE from the SVS-2 initial guess.

#### 3.1 SWE retrieval workflow





The workflow to retrieve SWE from Ku-Band SAR measurements is similar to what was presented by Pan et al. (2024), where snowpack variables are optimized iteratively using a Markov Chain Monte Carlo (MCMC) model to minimize the error between the simulated and measured  $\sigma^0$  (Section 3.5).

To initialize the snowpack variables, like the work of Singh et al. (2024), a land surface model was used to generate the prior distributions. In the context of an EO mission like TSMM, this allows for the prior distributions to evolve both spatially and temporally. In this study, the SVS-2 outputs of Woolley et al. (2024) were used to generate these prior distribution (Section 3.5.1). In order to improve computation efficiency, the multi-layered SVS-2 outputs were first reduce to two layers (Section 3.3). Since the simulations were done at point-scale for the TVC domain, all sites in Figure 1 were optimized using the same prior snowpack variable distributions. Both the default and Arctic versions of SVS-2 published by Woolley et al. (2024) will be investigated in this study, in order to determine the importance of defining more accurate snow priors to retrieve SWE with MCMC.

The MCMC method iteratively samples the snow variables (Section 3.5.3) from these prior distributions and converts them into  $\sigma^0$  using the Snow Radiative Transfer Model (SMRT) (Picard et al., 2018) model (Section 3.4). The probability of the sampled snow properties is then calculated using the likelihood function (Section 3.5.2) and the snow variable distributions are then updated to generate the posterior snow variable distributions. The posterior distributions are then compared to surveyed snow properties (Section 3.2) to assess the performance of the MCMC method. Since only single band and single polarization  $\sigma^0$  measurements were acquired for this TVC experiment, retrievals were done with measurements closest to the optimal incidence angle of 35  $^o$  (King et al., 2018). An extra test including four measurements in the proposed incidence angle range of TSMM ( $20^o < \theta < 50^o$ ) was conducted. Lower incidence angles being less sensitive to snow volume scattering and higher incidence angles being more sensitive to snow volume scattering, this emulates, without exactly reproducing, the dual Ku-Band frequency, dual polarization concept of TSMM.

#### 3.2 Reducing the in situ snow profiles to two layers

The snow profiles used in this study were presented in Montpetit et al. (2024) where detailed stratigraphy was surveyed during the TVC experiment and the measured snow profiles were reduced to two layered snowpacks. The methodology to obtain these reduced snowpacks is summarized here.

In order to have a representative snowpack at the 100 m spatial scale, scale at which the UMASS airborne Ku-Band SAR data has been processed (section 2.1), all the MagnaProbe snow depths, SMP density and SSA profiles, and complete snowpit measurements (temperature, snow cutter density, IceCube SSA and visual profile inspection) were used. The  $\sim$  290 MagnaProbe measurements per site were used to generate a snow depth distribution and its median value was used as its total snow depth. The SMP data measured behind the snowpit wall (2 to 3 measurements) with a vertical resolution of 2.5 mm was then calibrated (Proksch et al., 2015; Montpetit et al., 2024) into density and SSA profiles using the surveyed measurements from the density cutters and the IceCube instrument for SSA. Then, the 5 cm aggregated SMP profiles, thickness determined to

be small enough to represent average snowpack layers (Sandells et al., 2022; Montpetit et al., 2024), were classified into two grain types using the support vector machine methodology initially developed by King et al. (2020) and adapted to the 2018/19 TVC experiment by Montpetit et al. (2024): 1. rounded grains (R) or, 2. depth hoar (DH). From the classified SMP profiles, distributions of density and SSA were generated for the two snow layers. The median value of these distribution was then used as the density and SSA values for their corresponding snow layers. Finally, from the snowpit measurements, the median temperature measured for both snow layers was assigned to the representative snowpacks, even though temperature has little impact in the modelled backscattered signal for dry snowpacks (Picard et al., 2018), and was not considered in the MCMC optimization. Examples of representative snowpits are shown in section 4.1.

## 195 3.3 Reducing the SVS-2 snow profiles to two layers




SVS-2 can generate snow profiles of up to 50 layers. This considerably impacts the computation time of radiative transfer modelling using SMRT, thus increases the computation resources required to efficiently retrieve SWE using the MCMC approach. Meloche et al. (2025) have developed an objective method using K-means clustering in the extinction coefficient ( $k_e$ ) and layer height space, that generates a microwave equivalent snowpack from a multi-layered snowpack, that preserves the snowpack radiative transfer properties while retaining the bulk physical snow properties of the snowpack like SWE. They have shown that this approach can improve computation time up to 87 % when comparing SMRT simulations with a 50-layer snowpack and the equivalent 2-layer Microwave Equivalent Snowpack (MES).

For this study, all 120 ensemble members of both the default and Arctic version of the SVS-2 20-layer profiles were reduced to 2-layer using the Meloche et al. (2025) method. To do so, the  $k_e$ , calculated from the SVS-2 outputs using sub-modules of SMRT, in addition to the layer heights were used to classify the multi-layered snowpacks into two-layered microwave equivalent snowpacks. Examples of representative snowpits are shown in section 4.1. These two-layered snowpacks were then used to calculate the prior distributions used as first guesses for both snow layers into the MCMC method.

#### 3.4 Radiative transfer modelling

To convert these snowpack variables into simulated  $\sigma^0$ , the SMRT model is used (Picard et al., 2018). Similarly to Pan et al. (2024), the Improved Born Approximation (IBA) model is used to calculate snow scattering, which is implemented in the python open-source code of SMRT. The same radiative transfer modelling configuration used in Montpetit et al. (2024) is used in this study. Since the proposed TSMM SWE algorithm decouples the contributions to the measured  $\sigma^0$  from the soil and snow in a two step process, the soil properties retrieved by Montpetit et al. (2024), using lower frequency satellite SAR data, was used directly in SMRT. This is one of the difference with the methodology of Pan et al. (2024). Also, snow temperature was not optimized in this retrieval since it is known to have little impact on simulated  $\sigma^0$  of dry snow (Picard et al., 2018). The measured temperatures of both layers were assigned in SMRT since it is a required input to simulated  $\sigma^0$ .

#### 3.5 Markov Chain Monte Carlo method







The Markov Chain Monte Carlo (MCMC) method used for this SWE retrieval algorithm is coded using the open-source PyMC v5.16.2 python library (Salvatier et al., 2016; Abril-Pla et al., 2023), and was run on a high performance computing Linux cluster, hosted at ECCC. The architecture of the MCMC method was inspired from the work of Pan et al. (2024), but many aspects of the methodology used in this study are different and will be described in this section.

The MCMC method was initially run for 15 000 iterations. A portion of these iterations were used as the burn-in period (a maximum of 5 000 burn-in iterations was tested), e.g. these iterations are used to initialize the model and allow the sampling of the different variables to stabilize to values more representative of the observations. This burn-in period is not included in the iterations used to build the posterior distributions. Since MCMC tends to have correlated sampled variables between iterations, usually a large number of iterations is needed. Here, a maximum of 10 000 iterations were used. The Equivalent Sample Size (ESS), is an index that determines the number of samples that are uncorrelated (Martin et al., 2021), and helps to determine if the number of total iterations are sufficient. Additionally to the number of iterations, the MCMC method can use chains that are run in parallel. This ensures that the final posterior distributions converge to a similar solution for all chains. The chain convergence coefficient ( $\hat{R}$ ) is an index that calculate the between chain convergence of the posterior distributions (Gelman and Rubin, 1992). Both indexes are thus used to calculate the optimal number of iterations and chains to use. Tests were conducted in order to determine the optimal number of burn-in iterations, total iterations and chains needed to converge to proper solutions. These results are presented in section 4.2.

## 3.5.1 Prior distributions of snow properties

Similarly to the study of Pan et al. (2024), the initial prior distributions used as the first estimate of snow properties are constrained normal distributions. In order to make a methodology that works for all climates and all seasons, in the context of a satellite mission like TSMM, the means and standard deviations used to initialize these priors come from an ensemble of SVS-2 outputs. Since MCMC outputs are very sensitive to initial prior estimates, using a dynamic prior that changes through time and space allows for a more precise prior, which will result in a more precise SWE posterior estimate from the MCMC approach. This will be further discussed in section 5.2. Table 1 shows the means and standard deviations of all 120 members of both the default and Arctic versions of the SVS-2 outputs (Woolley et al., 2024). For snow height ( $H_{\text{snow}}$ ), the minimum value was chosen as the thinnest representative thickness of a layer (Sandells et al., 2022; Montpetit et al., 2024) and the maximum value was randomly put to 1 m even though no 1 m snowpack was measured at TVC during the campaign (e.g., see Figure 4 of Montpetit et al., 2024). The minimum and maximum values for snow density ( $\rho_{\text{snow}}$ ) and SSA were extrapolated from all the measurements of the field campaign. The values for SWE are also shown in Table 1 for reference and discussion purposes, since SWE is the desired retrieved parameter of the study. Examples of these priors are presented in section 4.1. In order to assess the importance of using the best possible source of data to generate these priors, means and standard deviations from the top 30 ensemble members of the Woolley et al. (2024) datasets were used. The impacts of the accuracy of the initial snow property estimates on SWE retrieval are shown in sections 4.3 and 5.2. Given the known higher uncertainty of the SVS-2

SSA outputs for both versions of the model (Woolley et al., 2024), compared to density and thickness, tests were conducted to increase the standard deviation of SSA for both snow layers, to assess its impact on retrieved SWE.

**Table 1.** Values used for the truncated normal distributions of the MCMC priors using the 120 and top 30 members of both SVS-2 versions of Woolley et al. (2024). Min and max values were extracted from all surveys of the 2018/19 TVC experiment. For reference and discussion, mean and standard deviation for the layered SWE information is also given, though these values are not used in the MCMC priors since SWE is not an explicit input to the model. Std stands for standard deviation.

|                                         | Grain Type | All 120 members |        |         |        | Top 30 members |        |         |        |       |       |
|-----------------------------------------|------------|-----------------|--------|---------|--------|----------------|--------|---------|--------|-------|-------|
| Snow Property                           |            | Mean            |        | Std     |        | Mean           |        | Std     |        | Min   | Max   |
|                                         |            | Default         | Arctic | Default | Arctic | Default        | Arctic | Default | Arctic |       |       |
| H <sub>snow</sub> (cm)                  | R          | 29.8            | 27.1   | 7.7     | 10.0   | 24.3           | 28.5   | 2.8     | 9.5    | 5.0   | 100.0 |
|                                         | Н          | 15.4            | 13.7   | 4.1     | 8.3    | 15.8           | 13.3   | 2.9     | 7.9    | 5.0   | 100.0 |
| $\rho_{\rm snow}$ (kg·m <sup>-3</sup> ) | R          | 217.7           | 246.5  | 14.6    | 20.4   | 231.1          | 235.4  | 5.2     | 8.2    | 150.0 | 450.0 |
|                                         | Н          | 190.0           | 200.3  | 30.0    | 40.8   | 218.7          | 199.0  | 3.2     | 36.0   | 100.0 | 350.0 |
| SSA $(m^2 \cdot kg^{-1})$               | R          | 12.7            | 11.6   | 1.6     | 2.8    | 12.1           | 11.2   | 1.4     | 2.5    | 10.0  | 50.0  |
|                                         | Н          | 5.2             | 4.0    | 1.4     | 1.5    | 6.1            | 4.0    | 0.8     | 1.5    | 8.0   | 25.0  |
| SWE (mm)                                | R          | 64.2            | 66.2   | 13.3    | 23.5   | 56.1           | 67.2   | 6.7     | 22.2   | _     | _     |
|                                         | Н          | 29.6            | 30.6   | 10.1    | 24.0   | 34.7           | 29.0   | 6.6     | 22.7   | _     | _     |

Finally, similarly to Pan et al. (2024) and Picard et al. (2022b), not knowing exactly the accuracy of the measured radar signal and its uncertainty given the variability of snow/soil properties at the 100 m scale, an uncertainty parameter ( $\delta$ ) was given to the measured and simulated backscattered signals, which were described by a normal distribution centered at the measured  $\sigma^0$  and  $\delta$  as its standard deviation. This uncertainty parameter is then fed into the likelihood function. The  $\delta$  prior was initialized at the radiometric accuracy of the UMASS antenna of 1 dB, with an uncertainty of 0.5 dB.

#### 3.5.2 Likelihood function


In order to improve computation efficiency, the log-likelihood function was used between the measured and simulated  $\sigma^0$  (Leung, 2022), and is given by:

$$l(\sigma_{mes}^0, \delta | \sigma_{sim}^0) = -\frac{1}{2} \left( \frac{\sigma_{mes}^0 - \sigma_{sim}^0}{\sigma} \right)^2 - \ln(\sqrt{2\pi}) - \ln(\delta)$$

$$(1)$$

where  $l(\sigma_{mes}^0, \delta | \sigma_{sim}^0)$  is the likelihood metric between the measured  $\sigma^0$  ( $\sigma_{mes}^0$ ) and simulated  $\sigma^0$  ( $\sigma_{sim}^0$ ), given an uncertainty on the measured  $\sigma^0$  ( $\delta$ ). This likelihood function is then used to calculate the Metropolis-Hastings likelihood ratio, which determines if the sampled snow parameters of the current iteration are accepted or rejected within the sampling strategy.

## 3.5.3 MCMC sampling

Given that the current version of SMRT uses the Discrete Ordinate Radiative Transfer (DORT) (Picard et al., 2018, 2013) method to solve the radiative transfer equation, and this solver is not differentiable for all variables, more modern and efficient samplers, like the No-U-Turn Sampler (NUTS) (Hoffman and Gelman, 2011), could not be used. This is why the Adaptive Differential Evolution Metropolis (DEMCZ) sampling (ter Braak and Vrugt, 2008) method, implemented in PyMC, was used in this study. This method differs from the original differential evolution metropolis (DEMC) sampling method (ter Braak, 2006) 270 since it uses information from past iterations to generate future jumps in sampled snow properties. DEMCZ also requires a lower number of chains (N) to be run in parallel in order to converge to a solution compared to N=2d, where d is the number of snow parameters to optimize, e.g. 3 snow parameters ( $H_{\text{snow}}$ ,  $\rho_{\text{snow}}$ , SSA) per layer, for a total of 12 chains for our current two-layer snowpack configuration. Also, for our specific SWE retrieval algorithm, N can be kept constant, where if we have more or less snow layers, we do not need to adapt the number of chains to run, even though the number of snow properties 275 change. DEMCZ is also known to be more efficient than random walk samplers. In this study, different number of chains were tested (a minimum of 4 and up to 12), and a total of 7 chains was chosen, in order to ensure proper sampling and good convergence (ESS and  $\hat{R}$ ), without compromising the computation efficiency of the algorithm. Similar results were obtained using 4 chains, but the model was less stable. The maximum number of chains was chosen as computation efficiency was not impacted, with the similar convergence, and model stability was preserved no matter the number of snow layers.

Also, since this experiment optimizes many snow variables and many combinations of these variables can provide the same simulated  $\sigma^0$ , constraints between layers for each variables were introduced, similarly to Picard et al. (2022a), where they constrained density profile to have a positive gradient with depth. Here, given the two-layer experiment, these constraints were determined based on local and published knowledge of the vertical profiles (see Figure 3 to 5). Hard constraints were put on density, SSA and thickness between the layers. If those constraints were not met, the sampled values for these three parameters were rejected. The density of the R layer had to be higher than for the DH layer. The thickness of the R layer was also constrained to be lower than the DH layer. Finally the SSA of the R layer had to be higher than the DH layer. The impact of these constraints will be presented in Section 4.4 and discussed in Section 5.

### 4 Results




In this section, the different sources of snow information will be presented in order to understand how the SWE retrieval algorithm is impacted within the MCMC method. The results of the tests to determine the MCMC parameterization will then be presented. Results of the SWE retrievals using the UMASS Ku-band SAR data will then be shown, and finally the snow posterior distributions will be compared to the measured in situ snow properties for different MCMC configurations.

## 4.1 Modelled and measured snow properties

Figure 3 shows the layer thicknesses (H<sub>snow</sub>) distributions for both SVS-2 versions and the measured layer thicknesses for the R (a) and DH (b) layers for all sites of the TVC experiment. The truncated normal distributions (Table 1) are overlaid on top of the histograms and the normal distribution of the measurements is shown in red and would consist in the idealized posterior distribution that the MCMC method would retrieve. We see that both versions of SVS-2 overestimate the thickness of the R layer and underestimate the thickness of the DH layer. That said, both versions show good overlap between their normal distribution and the idealized posterior distribution, suggesting that the MCMC method could converge to the proper thicknesses efficiently.

**Figure 3.** Measured snow thickness (H<sub>snow</sub>) distributions during the January TVC campaign, compared to the thickness distributions provided by the 120 SVS-2 ensemble members (Woolley et al., 2024) for the two dominant snow grain type layers a) rounded grains, b) depth hoar, and c) the total thickness, including both snow layers. Magenta lines represent the truncated normal distributions (Table 1) used as priors, using the mean and standard deviations of the two different SVS-2 versions. The red line represents the normal distribution, using the mean and standard deviation of the measurements, and consists in the desired posterior distributions obtained by the MCMC approach.

Figure 4 shows the same distributions as figure 3, for densities. Both SVS-2 versions underestimate the density of the R layer, and the Arctic version shows a better overlap with the idealized posterior distribution. The distribution of both SVS-2 versions overlap well with the idealized posterior distribution for the DH layer even though there is a tendency to slightly underestimate the density.

**Figure 4.** Same as Figure 3 for snow density ( $\rho_{\text{snow}}$ ).

Figure 5 shows the same distributions for SSA. Here, it is clear that SVS-2, no matter the version, underestimates the SSA and even outputs values that are below the minimum values measured in the field. In order to achieve the overlap shown in Figure 5, the SVS-2 standard deviations had to be tripled.

Figure 6 shows the same distributions for SWE. We see that SVS-2 tends to overestimate SWE for the R layer and underestimate SWE for the DH layer. Variability in modelled layered SWE for both versions of SVS-2 is similar to what is observed in the field. This results in a very narrow range of modelled bulk SWE that fit very well with the observations. This tends to indicate that SVS-2 reproduces the bulk SWE properly over TVC but has more difficulty in properly representing SWE stratigraphy. Figure 6c also shows that the higher uncertainties and the over- and underestimations of the layered SWE tend to cancel out for the bulk properties. It should be noted that the distributions in Figure 6c do not realistically represent the priors and the uncertainty on SWE since SWE is not an explicit variable used in the MCMC model.

Impacts of the different prior distributions shown in Figures 3, 4, and 5 will be discussed in section 5 in the context of the SWE retrieval.

## 4.2 MCMC algorithm parameterization



To determine the optimal MCMC parameterization, different tests with different numbers of iterations were run. It was determined (not shown) that beyond 1000 burn-in iterations (a total of 5000 burn-in iterations were tested), no significant improvement was observed to converge to a proper solution. Below 1000 burn-in iterations, more chains in parallel and a larger number of total iterations were needed for the method to converge, but a larger uncertainty on the posterior distributions was observed.

Figure 5. Same as Figures 3 and 4 for SSA. Since SSA is grain type specific, a bulk snowpack value is not shown, as in Figures 3 and 4.

Figure 6. Same as Figures 3 and 4 for SWE.

With 1000 burn-in iterations, a test with 10 000 iterations was run over all sites to determine the optimal number of iterations necessary. Figure 7 a) shows the  $\hat{R}$  (Gelman and Rubin, 1992) over all the iterations. The mean, minimum and maximum  $\hat{R}$  for

all the sites are shown in Figure 7 a), as well as the recommended acceptable threshold (Vehtari et al., 2021). It it shown that after 4000 iterations, all  $\hat{R}$  values are below 1.1 and some start to be below the recommended threshold, which is considered acceptable in certain contexts, where the ESS is large enough.

Figure 7 b) shows the evolution of the ESS for the 10 000 iterations. We see that after 5000 iterations, all ESS values are beyond the acceptable threshold of 100 and are five times greater on average. These results, with the results of Figure 7 a) indicate that the optimal number of iterations is around 5000 iterations. Figure 7 c) shows the evolution of the SWE RMSE over all the iterations. We see that after 4000 iterations, no considerable gain is achieved in retrieving SWE. With the results shown in these three figures, 5000 iterations was selected in order to ensure optimal SWE retrieval and proper convergence of all the retrieved snow properties for all snow layers. All the results of the following sections were thus obtained after 5000 iterations.

## 4.3 MCMC retrieved snow water equivalent








The following results present the impact of using different priors in the MCMC method on the retrieved SWE. Figure 8 shows the retrieved SWE from the MCMC method with priors coming from all 120 ensemble members of the default and Arctic SVS-2 versions (Woolley et al., 2024). The truncated normal distributions used for the priors were generated with the mean values and the standard deviations (Table 1). The original modelled SWE values from both SVS-2 versions are shown in red, with the variability in modelled bulk SWE among the 120 ensemble members is shown in the red shaded area. The expected SWE retrieval accuracy of the TSMM mission, for an Arctic snowpack (30 mm, see Derksen et al., 2019), is also shown in this figure. Retrieved SWE from the default SVS-2 priors show larger RMSE (27.6 mm) than the ones retrieved with the Arctic SVS-2 priors (20.9 mm). These results are summarized in Table 2. Retrieved SWE uncertainty is also shown in Table 2. Here, uncertainty is defined as quartile deviation instead of the usual standard deviation since the posterior distributions are not strictly normal distributions. Little variability in the retrieved SWE from both SVS-2 versions can be observed. There is also an offset between the retrieved SWE and the original SVS-2 modelled SWE. The uncertainty on the retrieved SWE values (error bars) is slightly better for the default version of SVS-2 (11.9 mm mean quartile deviation) compared to its Arctic version (19.6 mm mean standard deviation). Nonetheless, the SWE estimates from the Arctic version show only two points outside the TSMM expected accuracy compared to eight points for the default version. Since the expected TSMM accuracy is on the RMSE criterion, both tests meet the requirement.

We see that the initial modelled SWE value of the Arctic version of SVS-2 is slightly better than the one from the default version. The Arctic version also shows little variability (red shaded area) compared to the default version, in its initial SWE estimate. Both versions of the models do not represent the range of SWE values that were measured in the field. These observations will be discussed in section 5, supported by results shown in section 4.4. Knowing that many of the ensemble members of both SVS-2 versions were not representative of the Arctic snowpack, the same test was processed using the top 30 ensemble members, which were determined to be more representative of snowpacks surveyed at TVC (Woolley et al., 2024).

Figure 9 shows the retrieved SWE using the top 30 SVS-2 ensemble members as priors. The default version of SVS-2 seem to provide better results (RMSE = 17.9 mm) in terms of SWE retrieval. The values are in fact close to the original modelled

Figure 7. Evolution of a)  $\hat{R}$  (Gelman and Rubin, 1992), b) Equivalent Sample Size (ESS) (Martin et al., 2021) for all variables, all sites, and seven chains; and the SWE RMSE over the 10,000 iterations of the MCMC optimization. Acceptable  $\hat{R}$  threshold published by Vehtari et al. (2021), and ESS threshold published by Kass et al. (1998) are shown in red.

SVS-2 SWE. There is still little variability compared to measured SWE values. The Arctic version shows lesser performances (RMSE = 21.2 mm), with a similar offset shown in figure 8, but the retrieved SWE show a bit more spread. The initial SVS-2 SWE values do not differ considerably from the previous test. One significant result compared to past tests is the uncertainty of

**Figure 8.** Comparison of the retrieved SWE using the MCMC approach with priors coming from all 120 ensemble members of the a) default and b) Arctic versions of SVS-2 (Woolley et al., 2024). The error bars show the 1st and 3rd quartiles of the measured (x-axis) and posterior (y-axis) distributions. The red shaded areas show the 1st and 3rd quartiles of the SVS-2 distributions (Figure 6c).

the retrieved SWE values (errors bars) for the default version are much narrower (6.9 mm mean quartile deviation). From the past two figures, the top 30 default SVS-2 ensemble members seem to perform best, where most points and their uncertainty fit within the expected accuracy of TSMM. However, with results presented in section 4.4, the default version was rejected for the following tests, due to the retrieved posterior snow properties (see section 4.4).

Figure 10 shows the results of the SWE retrieval when increasing the uncertainty on the SSA ( $\delta_{SSA}$ ), and when including four radar observations from different angles. When comparing the impact of increased  $\delta_{SSA}$  on priors, we see that the overall accuracy is improved, with an RMSE = 18.7 mm, compared to 20.9 mm (Figure 8). Similar spread can be observed, but one observed improvement is the lower difference between the retrieved SWE values and the original SVS-2 modelled SWE. One interesting result to note is that the uncertainty (22.5 mm quartile deviation) on the retrieved SWE values (errors bars) are not considerably impacted compared to the original Arctic test (19.6 mm quartile deviation).


The greatest improvement can be observed when retrieving SWE using four  $\sigma^0$  measurements. The lowest RMSE was obtained (15.8 mm), out of all the tests, and all values are within the expected accuracy of the TSMM mission. Again, the uncertainty (23.4 mm quartile deviation) on the retrieved SWE values (error bars) were not considerably impacted, though in some cases, the uncertainties are slightly larger.

**Figure 9.** Comparison of the retrieved SWE using the MCMC approach with priors coming from the top 30 ensemble members of the default and Arctic versions of SVS-2 (Woolley et al., 2024). The error bars show the 1st and 3rd quartiles of the measured (x-axis) and posterior (y-axis) distributions. The red shaded areas show the 1st and 3rd quartiles of the SVS-2 distributions (Figure 6c).

**Table 2.** Retrieved SWE RMSE and uncertainty, for different parameterizations of the MCMC method. Uncertainty is defined as the quartile deviation, and the values consist in the mean value over all sites. Percentages over the mean surveyed SWE values are given in parentheses. Tests were conducted with the default SVS-2 version as priors and larger  $\delta_{SSA}$  but due to MCMC convergence issues, results are not shown here.

| Parameterization              | SWE RM       | ISE (mm)     | SWE uncertainty (mm) |              |  |  |
|-------------------------------|--------------|--------------|----------------------|--------------|--|--|
| Parameterization              | Default      | Arctic       | Default              | Arctic       |  |  |
| 120 ensemble members          | 27.6 (28.7%) | 20.9 (21.7%) | 11.9 (12.9%)         | 19.6 (21.1%) |  |  |
| Top 30 ensemble members       | 17.9 (18.7%) | 21.2 (22.0%) | 6.9 (7.4%)           | 18.8 (20.2%) |  |  |
| Larger $\delta_{\rm SSA}$     | _            | 18.7 (19.4%) | _                    | 22.5 (24.2%) |  |  |
| Larger $\delta_{SSA}$ + 4 obs | _            | 15.8 (16.4%) | _                    | 23.4 (25.2%) |  |  |

#### 375 4.4 MCMC retrieved snow properties

In this section, the impacts of prior selection and constraining valid snowpack properties, between layers, within the MCMC method will be presented. The SM site (Figure 1) was chosen as an example for these results, and is typically what is observed for all sites. Figure 11 shows posterior distributions of snow parameters from the MCMC method and the normal distribution of the surveyed snow measurements (snowpits) and priors from the default SVS-2 version (column a), the Arctic SVS-2 version

Figure 10. Comparison of the retrieved SWE using the MCMC approach with priors coming from a) the top 30 ensemble members of the Arctic versions of SVS-2 (Woolley et al., 2024) and b) using the same top 30 ensemble members of the Arctic SVS-2, tripled prior uncertainty on the SSA ( $\delta_{SSA}$ ) and three additional  $\sigma^0$  observations (different incidence angles), emulating the number of observations the TSMM would acquire. The error bars show the 1st and 3rd quartiles of the measured (x-axis) and posterior (y-axis) distributions. The red shaded areas show the 1st and 3rd quartiles of the SVS-2 distributions (Figure 6c).

(column b) and the default SVS-2 version without constraining the valid snow properties (column c). The evolution of the MCMC sampling for all snow parameters are also shown throughout the 5000 iterations in Figure 12, for the three same scenarios.




Figure 11 a) shows that, using the default version of SVS-2, SWE tends to be underestimated for this site compared to its initial prior estimation, e.g. the bias between medians is 24.5 mm higher for the posterior than the prior, compared to the snowpit distributions. Posterior thicknesses for both layers show improvement from their respective priors, i.e. an improvement of 14.4 cm and 3.8 cm for the R and DH layers respectively. Densities show little improvements even after 5000 iterations. Posterior and prior distributions tend to overlap well, without much convergence towards the snowpit distributions, with only a difference of 4.9 kg  $\cdot$  m<sup>-3</sup> and 0.1 kg  $\cdot$  m<sup>-3</sup> difference between their medians for the R and DH snow layers respectively. SSA for DH shows improvement where the posterior median is closer to prior median by 0.7 m<sup>2</sup>  $\cdot$  kg<sup>-3</sup>. There is an improvement of the SSA for the R layer but tend to still be largely underestimated compared to measured SSA, with a bias of -20.4 m<sup>2</sup>  $\cdot$  kg<sup>-3</sup>.

Figure 11 b) shows that, using the Arctic version of SVS-2, SWE is slightly more underestimate by the posterior compared to the prior, i.e. the bias between medians is 24.7 mm higher for the posterior than its prior, which had a bias of 6.3 mm with the snowpits measurements. SWE estimate is better than what was estimated using the default SVS-2 version, where a bias of -18.5 mm is obtained for the Arctic SVS-2 prior compared to -36.1 mm for the default SVS-2 prior (Figure 11 a). Again,

thicknesses are well estimated by posteriors compared to priors, with 16.5 cm and 5.8 cm improvements on biases for the R and DH snow layers respectively when compared to measurements. This shows that both sources of priors tend to perform well as first guesses for the MCMC method. Density posteriors still show some differences with measurements (-50.3 kg  $\cdot$  m<sup>-3</sup> and -31.1 kg  $\cdot$  m<sup>-3</sup>biases for R and DH snow layers respectively) but the errors are considerably lower than the estimates from the default SVS-2 version, where improvements of 74.0 kg  $\cdot$  m<sup>-3</sup> and 10.8 kg  $\cdot$  m<sup>-3</sup> on biases were observed for the R and DH snow layers. The same observation can be made for the SSA posteriors, where an improvement of 11.8 m<sup>2</sup>  $\cdot$  kg<sup>-3</sup> and 4.7 m<sup>2</sup>  $\cdot$  kg<sup>-3</sup> is observed compared to results of figure 11 a.

Figure 11 c) shows the same results when no constraints are given to the sampled snow parameters between layers, and is mainly presented for discussion purposes in section 5. The best results in terms of SWE estimate is observed compared to results of Figures 11 a) and 11 b), with a bias of -8.4 mm. Nonetheless, the thicknesses show the worst estimates (bias of 14.9 cm and -9.0 cm for R and DH grain layers) and do not deviate from the prior estimates (0.1 cm and 1.1 cm difference between the posterior and prior medians for R and DH grain layers). The same observation is made for the density of the R layer with a bias of -127.6 kg · m<sup>-3</sup> for the posterior compared to -129.5 kg · m<sup>-3</sup> for the prior. The density of the DH layer is well estimated though (bias=0.3 kg · m<sup>-3</sup>) and shows the best results out of all MCMC estimates. Similar observations can be made for the SSA where only a slight improvement is seen for the R layer (bias=-188.2 m<sup>2</sup> · kg<sup>-3</sup>), and excellent estimation of the DH layer SSA is seen (bias=0.2 m<sup>2</sup> · kg<sup>-3</sup>).

Figure 12 shows the sampling evolution of the MCMC method for the same three scenarios as in Figure 11. We see that with less observations (one observations for Figures 12 a and c, and four observations for Figure 12 b), the sampling converges more rapidly and shows less variability, which is also shown in the spread of the retrieved parameters (Figure 11). One parameter that does not show as much variability with more observations is the radar  $\sigma^0$  uncertainty ( $\delta$ ). The variability of the  $\sigma^0$  measurement tends to converge around 1.1 dB  $\pm 0.3$  dB, 1.2  $\pm 0.2$  dB and 1.2  $\pm 0.4$  dB for Figures 12 a), b) and c) respectively. These results also show that with poor prior estimates (Figure 12 a) and an unconstrained optimization (Figure 12 c), some snow parameters, especially for the R layer quickly converge to a poorly estimated value, even though the SWE estimate is close to measurements.

The differences in snow profile estimates between the different selected prior distributions and MCMC parameterizations will be further discussed in the next section. The implication for different applications will also be discussed.

#### 420 5 Discussion



#### 5.1 Retrieving SWE with MCMC

Results of section 4 show that, like previous studies (Pan et al., 2024), the MCMC method is very powerful to fit, in this case, SAR  $\sigma^0$  observations with modelled  $\sigma^0$ , when many variables need to be optimized simultaneously. Figure 11 shows that, without proper constraints, the MCMC method can use the  $\sigma^0$  information to optimize snow parameters and still achieve great results when comparing to bulk SWE, but returns a snow profile that is not representative of what is found in the field (Figure 11 c). These results confirm that SAR  $\sigma^0$  is sensitive to SWE in the Ku-Band range, since, even with a poorly estimated microstructure (Figures 11 a and c), which is an important parameter that drives snow volume scattering in that frequency range

(Montpetit et al., 2024; Picard et al., 2022b; King et al., 2018), other variables like thickness are tuned to fit the measured  $\sigma^0$ (Figure 12 c), and can still achieve a reasonable SWE estimate compared to measurements. It should be noted that when SWE is poorly estimated by the prior, the posterior SWE estimate has a higher error (Figure 9), where SWE estimates are concentrated around the initial modelled SWE and do not diverge from that initial estimate. Also, further tests were done (not shown), where the uncertainty on the SWE was increased, by increasing uncertainty on thickness and density individually and separately. Every tests resulted in underestimation of SWE, most likely due to the underestimation of SSA in the priors, which boosted the volume scattering of both layers. The most sensitive parameter in the MCMC model being thickness, it reduced the snow thickness to reduce the volume scattering and fit the modelled  $\sigma^0$  with the measured  $\sigma^0$ , resulting in an underestimation of SWE. Figure 11 c) also confirms previous observations (King et al., 2018) that Ku-Band  $\sigma^0$  is most sensitive to the DH layer rather than the R grain wind slab layer. Parameters from the DH layer show lower median biases than the R layer, and the latter also tends to stick to its prior distribution, indicating lower sensitivity of the  $\sigma^0$  to the R layer. It explains why the posterior SWE estimates are lower than the initial SVS-2 estimates for both versions (Figure 8), since  $\sigma^0$  is very sensitive to both SWE and microstructure. The thickness of the R layer is properly estimated, i.e. MCMC reduces its thickness to lower the scattering caused by the low SSA estimation, and does not increase its density sufficiently to properly estimate SWE. The fact the SVS-2 underestimates the R layer density, and that the MCMC model struggles to sample values that are close to measured densities, aggravates the underestimation of SWE. This is why it is important to have some knowledge of stratigraphic snow properties, e.g. number of snow layers, density and SSA gradients, to constrain the MCMC method to valid snow properties without overfitting on the most sensitive parameters, and to not over-trust the initial prior estimates, i.e. not be too restrictive on the prior uncertainties. With inter-layer constraints (Figure 11 a and b), it is possible to achieve SWE estimates within desired errors, like the 30 mm RMSE determined for the TSMM mission (Derksen et al., 2019), and comparable to the unconstrained MCMC method. A way to potentially solve the issue of the high sensitivity to layer thickness would be to use SWE as a prior directly and infer snow density and thickness from published relationships between SSA and density (Domine et al., 2007). This will be further tested in future experiments in the context of TSMM.






The impact of the initial guess is also valid for other snow parameters. There is a very fine balance to identify between prior estimates and their uncertainties. The farther the initial guess is from the ground truth, higher is the number of iterations needed for MCMC to converge towards a final solution (Pan et al., 2024). Also, increasing the uncertainty on the priors tend to increase the uncertainty on the posteriors and a larger number of iterations is also needed to converge to a solution. Figure 10 shows that by increasing the uncertainty on SSA, a known snow parameter to be highly underestimated by SVS-2 (Woolley et al., 2024), the accuracy of the retrieved SWE is improved. By allowing the method to sample SSA in a wider range of possible values, closer to what was measured in the field, more weight is given to snow microstructure in the modelled  $\sigma^0$ . In this case, increasing the uncertainty on SSA does not directly impact the uncertainty on retrieved SWE, since SWE is a function of density and thickness. Nonetheless, it does have an impact on the  $\sigma^0$  measurement uncertainty ( $\delta$ ), which indirectly adds uncertainty to all snow variables.

Similarly, when comparing the outputs from both SVS-2 versions, the prior density estimates for the R layer of the default version (Figure 11 a), do not allow to sample values close to the measured  $\rho_{\text{snow}}$ , due to the lower prior uncertainty, which

prevents the MCMC method to properly sample other variables, such as SSA for the same layer, since volume scattering in the IBA model depends on both SSA and density. It should be noted from Figure 11 a), that a secondary peak had started to form, for both  $\rho_{\text{snow}}$  and SSA for the R layer, closer to the measured values after 5000 iterations. This indicates that with a higher number of iterations, it is possible that the method could have converged towards a better solution, even with a less accurate first guess, showing the potential of MCMC to retrieve snow parameters. That said, a test was conducted to confirm this hypothesis (not shown here), with 40,000 iterations and no significant improvement was observed compared to Figure 11 a). With the initial estimate of  $\rho_{\text{snow}}$  being closer to what was measured with the Arctic version of SVS-2, we clearly see that after 5000 iterations, the method converges towards a solution that is closer to the measurements for all snow parameters. Figure 7 c) also shows, that increasing the number of iterations does not improve SWE retrieval.





As shown in Figure 10, the largest gain in SWE accuracy comes from adding more observations to the retrieval method. In this study, measurements at different incidence angles were available, which modified the sensitivity of the  $\sigma^0$  to the different scattering mechanism (Tsang and Kong, 2001), thus modifying the importance of the different snow parameters in the retrieval process. This could explain why the retrieved uncertainty on the radar  $\sigma^0$  ( $\delta$ ) is less variable in Figure 12 b), since with four observations, that uncertainty is spread out over the snow parameters and less on the  $\sigma^0$  measurements. This result also shows that, even though the lower Ku-Band frequency (13.5 GHz) is sensitive to SWE and snow volume scattering, it still has a high sensitivity to surface scattering at the soil-snow interface, which could explain why there is not significant spread among the retrieved SWE. The lack of spread in the retrieved SWE can also be explained by the low uncertainty on the thickness priors of both layers, which are the most sensitive parameters in the MCMC model, but are restrained to a narrow range of thickness values. Also, the fact that volume scattering is less impacted by density, than it is by SSA, reduces the potential of sampling a wider range of SWE values, which is a function of snow thickness and density. The higher error coming from using optimized effective soil properties (increased RMSE of 0.4 dB) for all the TVC domain instead of site specific variables (see Fig.13 of Montpetit et al., 2024), propagates in the uncertainty of the  $\sigma^0$  measurement and impacts the retrieved snow properties. Since the uncertainty on the modelled  $\sigma^0$  values ( $\delta$ ) are of the order of 1.5 dB, compared to a change in RMSE of 0.4 dB, it is unlikely that considering site specific soil properties will have a significant impact on the retrieved SWE in this study. That said, it was shown by (Montpetit et al., 2024) that soil cannot be neglect for SWE retrievals at Ku-band, and that its properties must be properly estimated. This result confirms the choice of the dual frequency, dual polarization concept for the TSMM mission, where four observations will be made available for each satellite pass. The higher Ku-band frequency (17.25 GHz) being more sensitive to snow micro-structure, and the cross-polarization being more sensitive to volume scattering, especially from the DH layer (Ulaby and Ravaioli, 2020). This concept should allow the MCMC method to converge towards a good stratified snow profile estimate, given that proper stratigraphic information is known, i.e. layering, DH fraction, vertical density SSA gradients, etc. For this study, this knowledge was based on field observations, which is not possible to achieve operationally at the continental scale. This is why, improved snow modelling for different landscapes, and improve data assimilation schemes are necessary to enhance the predictability and assessment of these stratigraphic conditions.

## 5.2 Current limitations of snow physical models

Results of section 4.3 show that properly estimating the initial guess for the different snow properties is crucial to accurately retrieve SWE using the MCMC approach. In an operational context, computation efficiency is important. This is why a proper prior is important (Figure 9) to improve the accuracy of SWE estimates but also reduce the number of iterations needed for the MCMC method to converge to a solution. Again, in an operational context over various landscapes, as seen in Canada, it is important to rely on snow modelling such as what SVS-2 can provide, in order to spatialize the priors but also allow them to evolve in time, thus adapting the priors in both space and time.

That said, we have seen that the higher uncertainty on the SSA estimates (Figure 5) makes it challenging to use directly the SVS-2 estimates as priors. Same observation is true for the density of the R layer (Figure 4), where the density is strongly underestimated, making it more difficult for MCMC to converge to a realistic solution. These higher uncertainties mainly come from the fact that Crocus, the snow physical model implemented in SVS-2, was originally developed to simulate alpine snow. Ongoing work will implement new snow physical processes in Crocus, and improve the modelling of vertical physical processes for the different climates observed in Canada.

This study has shown that, even though the priors may have higher uncertainties, it is still possible to retrieve SWE within the 30 mm RMSE threshold (Figure 10) set for TSMM. The proposed improvements above, supported by results of section 4, should provide a more efficient and accurate retrieval algorithm that could be applied to a large and diverse landscape, such as Canada. This study is the initial step to creating a SWE retrieval algorithm that can be applied both spatially and temporally. The validation done here, in an Arctic environment, will be reproduced in other global climates, and will be used to further enhance the SWE retrieval algorithm.

#### 515 6 Conclusions






This study uses the previously published Trail Valley Creek (TVC) experiment 2018/19 dataset (Montpetit et al., 2024) in order to developed a snow water equivalent (SWE) retrieval method inspired by previous work using the Markov Chain Monte Carlo (MCMC) method (Pan et al., 2024). The heart of the retrieval algorithm relies on the Snow Radiative Transfer Model (SMRT) model (Picard et al., 2018) which allows to minimize a likelihood function between the measured and modelled backscatter measurements at Ku-band (13.25 GHz). Here, the measured  $\sigma^0$  come from the University of Massachusetts instrument mounted on board a Cessna-208. The retrieved SWE and layered snow properties from the MCMC method were compared with field measurements surveyed during the 2018/19 TVC experiment.

Compared to previous studies retrieving SWE under dry snow conditions using the MCMC method, here we neglect certain snow parameters like snow temperature, which do not have a significant impact on radar backscatter radiative transfer modeling (Picard et al., 2018). This study focuses on retrieving snow properties and uses the retrieved underlying soil properties needed for radiative transfer modelling from Montpetit et al. (2024). We also show that, in order to create an efficient SWE retrieval algorithm applicable to various climates and landscapes, the new version of the land surface model used in support of environmental forecasting at Environment and Climate Change Canada, Canadian Soil Vegetation Snow version 2 (SVS-2) (Woolley

et al., 2024; Garnaud et al., 2021; Vionnet et al., 2022) can be used to generate prior distributions for the MCMC method. This is crucial for future satellite missions such as the Terrestrial Snow Mass Mission (TSMM). Given the results shown in this study, we should also expect that by allowing the priors to evolve in time and space, given the dynamic seasonal evolution of weather and snow conditions, a reduced number of iterations will be needed for the MCMC method to converge to a solution, thus improving computation efficiency. Since the SWE retrieval algorithm optimizes  $3 \times N$  parameters simultaneously, where N is the number of snow layers present in the snowpack, we also implemented the snowpack layer reduction method published by Meloche et al. (2024) to improve computation efficiency, which reduces the number of layers to a relevant number, i.e. a rounded grains (R) wind slab snow layer with an underlying coarse depth hoar (DH) snow layer (Montpetit et al., 2024).

Even though the SVS-2 outputs do not reflect perfectly the measured snow height ( $H_{snow}$ ), snow density ( $\rho_{snow}$ ) and Specific Surface Area (SSA) during the 2018/19 TVC experiment (Section 4.1), it is possible to increase the uncertainty on the prior distributions for the known snow properties to have higher errors in order to retrieve SWE accurately (Section 4.3). These are also known limitations of the Crocus model for Arctic snowpacks and work is ongoing to improve the model to better represent modelled snow properties over various climates. This work also indicates that land surface models like SVS-2 and radar measurements can work together to mutually improve their accuracies. This is part of the TSMM concept where SVS-2 and the radar measurements will work together with a data assimilation scheme to mutually improve their estimates, particularly in remote regions with little observations (Derksen et al., 2019).

It was shown that it is important to have priors that reflect typical values observed in the field and to constrain the inter layer valid properties (e.g.  $SSA_R > SSA_{DH}$ ), since the MCMC tends to optimize parameters that influence the radar  $\sigma^0$  the most. This can lead to a better SWE estimate (Table 2) but with a very different retrieved vertical snow profile compared to measurements (Figure 11). This has significant impacts for many hydrological applications which require stratified snow properties (Cristea et al., 2022), and could also impact numerical prediction systems which uses retrieve snow properties in their data assimilation scheme (Alonso-González et al., 2022).

It was also shown that the best improvement to SWE accuracy and uncertainty was to include more  $\sigma^0$  observations, where the different observations are more or less sensitive to either surface or volume scattering. This was achieved here by including observations at various incidence angles. A SWE RMSE of 15.8 mm was achieved when including four observations and a larger uncertainty on SSA, allowing MCMC to more rapidly sample values included within the measured distributions. This result confirms assumptions used to develop EO missions to retrieve SWE such as CoReH2O (Rott et al., 2010), and TSMM (Derksen et al., 2019), where dual-frequency and dual-polarization concepts are put forward, giving four observations for a single satellite pass. The higher frequency in the dual-frequencies and the cross-polarization term ensures a higher sensitivity to snow volume where the lower frequency and the co-polarization term ensures a higher sensitivity to surface scattering properties such as the snow-soil interface.

Work is still required in order to operationalize SWE retrieval algorithms such as the one proposed in this study, but it confirms, along with previous studies (Singh et al., 2024; Pan et al., 2024; Durand et al., 2024; Lemmetyinen et al., 2022), the feasibility of such EO missions.

Data availability.

575

Code and data availability. All codes are available at https://github.com/ECCCBen/TVCExp18-19\_SWE. Links to the different datasets used are provided in the Github repository.

Author contributions. BM wrote the manuscript with contributions from all co-authors. The late Joshua King designed the experiment (see Montpetit et al. (2024)). PS, MA and his team at UMass developed the airborne radar and processed the data. BM performed the analysis. BM, JK, CD and PT collected the field measurements. MB helped write portions of code used and reviewed the codes before publication. VV, GW and NL provided the SVS-2 data, reviewed the manuscript and provided analysis guidance in the context of the TSMM mission.

570 Competing interests. Some authors are members of the editorial board of journal The Cryosphere.

Acknowledgements. This work was started and field campaign orchestrated by the late Joshua King. The study was completed by co-authors. Trail Valley Creek activities were supported by Environment and Climate Change Canada, the Canadian Space Agency and NASA's THP and ESTO-IIP programs (Grant numbers: 80NSSC20K1592, 80NSSC22K0279). The authors would like to thank the excellent logistical support provided by the Trail Valley Creek station crew, in particular Branden Walker and Philip Marsh. This work would not have been possible without the contribution of many partners including Arvids Silis (ECCC), Peter Toose (ECCC), Barum Majumber (WLU), Alexandre Roy (UQTR), Alex Mavrovic (UQTR), Daniel Kramer (UdS), Simon Levasseur (UdS), Casey Wolieffer (UMass), Nick Rutter (Northumbria U.), Richard Essery (Northumbria U.), Jim Hudgson (Lake Central Aircraft Services), Anna Wendleder (DLR), Yves Crevier (CSA) and Simon Yueh (NASA).

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

Figure 11. Example from the SM site of prior distributions coming from the default (column a, same prior parameterization as Figure 9a) and Arctic (column b, same prior configuration as Figure 10b) versions of SVS-2 and retrieved posterior distributions from the MCMC method, for the different snow variables compared to the surveyed snow measurements. Column c) consists in the MCMC optimization using the default version of SVS-2, where no vertical constraints on snow properties were applied.  $\delta$  is the free parameter corresponding to the uncertainty of the radar backscatter measurement and was not measured in the field.

**Figure 12.** Evolution of the MCMC sampling for the 5000 iterations which correspond to the posterior distributions of Figure 11 for the same three optimization scenarios. Full horizontal lines consist in median values from measurements (snowpits) and the dashed horizontal lines consist in the mean of the priors. The min/max values consist in the minimum and maximum properties sampled at each iteration between the seven parallel sampling chains.