# Peer review of "Snow Water Equivalent from airborne Ku-band data: The Trail Valley Creek 2018/19 Snow Experiment"

_EGUsphere, 2025_

## Author Comment (AC1)

**Authors Response to Reviewers**

Montpetit et al.

**Correspondence:** Benoit Montpetit (benoit.montpetit@ec.gc.ca)

**1 Response to reviewer # 1**
**by: Anonymous Reviewer**

**1.1 General Comments**

In this paper, the authors expanded an originally developed Bayesian-based SWE estimation algorithm to a new Ku-band radar
sensor applied in Canada. Several important improvements were made.

The achieved SWE estimation accuracy was below 20 mm, and could be improved to 15.8 mm if three additional observation angles are provided.

The authors made great efforts to explicitly describe the influence of the prior mean and variance on the accuracy of the re-trieval results and their MCMC-estimated chain uncertainties. They also described the ability of MCMC and a proper constraint
setting to correctly characterize a layered snow stratigraphy. The discussions are in-depth, and most of them are correct.

I have only the following suggestions to post.

The authors would like to thank the reviewer for the great positive feedback provided which considerably improves the manuscript. By answering the major comments below, many details in the SVS-2 model data used and Section 2.3 were greatly improved. Figure 1 was also modified with the suggestions of the reviewer.

### 1.2 Major comments

1. In abstract, what is the physical snow RT model utilized to describe the backscattering in four incidence angles?

The following text was added in the abstract:
*... coupled with the Snow Microwave Radiative Transfer model (SMRT) ...*

2. It is suggested to provide a false-color image, a DEM, and a land cover in addition to the backscatter image in Figure 1.

Figure 1 was redone to include suggestions from both reviewers. Revised legend:

*Figure 1. Sites sampled during the January campaign of the Trail Valley Creek (TVC) 2018/19 experiment. Squares correspond to a 100 m x 100 m around the central surveyed snowpit (see Section 2.2). Background images are two overlapped University of Massachusetts (UMASS) Ku-Band radar images corresponding to two different flight passes acquired November 14, 2018 (left, Siqueira et al., 2021), the 2 m ArcticDEM (center, Porter et al., 2023), and the vegetation classification (right, Grünberg*
*and Boike, 2019).*

3. For Section 2.3, it reads unclear whether the SVS-2 simulation dataset is a full region map or one that covers only several individual points. Additionally, the description of the forcing dataset contradicts itself between line 126 (SM in Figure 1) and line 131 (neighboring weather stations).

This has been clarified to indicate that the simulations were done at point-scale at the main weather station (SM). Sentence
at L. 131 was removed and information was added to sentence at L. 126 which now reads:

*These outputs were generated from point-scale simulations located at the main meteorological site of TVC (SM site in Figure 1), where most of the meteorological forcing data was acquired, and complemented by neighbouring stations when data was not available.*

4. Can SVS-2 consider wind compaction and effectively model the wind slab layer for snow in Canada?

This was the objective of the study of Woolley et al. (2024), to retrieve a more realistic density and SSA profile, as seen in the Arctic. Though the density of the wind slab layer is still underestimated, the observed vertical density gradient, i.e. higher density at the surface and lower density at the bottom (depth hoar), is now better represented in the Arctic version of SVS-2. Details are now added in Section 2.3, and is further discussion in Section 5, where the underestimation of the density of the wind compacted layer by SVS-2 introduces challenges for the SWE retrieval.

*Section 2.3: Both versions used in the study of Woolley et al. (2024) are tested in this study, where an Arctic version of Canadian Soil Vegetation Snow version 2 (SVS-2) was developed to improve the overall snow properties and stratigraphy of Arctic snowpacks.*

*Section 5: The thickness of the rounded grains (R) layer is properly estimated, i.e. Markov Chain Monte Carlo (MCMC) reduces its thickness to lower the scattering caused by the low Specific Surface Area (SSA) estimation, and does not increase its density*
*sufficiently to properly estimate snow water equivalent (SWE). The fact the SVS-2 underestimates the R layer density, and that the MCMC model struggles to sample values that are close to measured densities, aggravates the underestimation of SWE.*

5. For lines 147-153, does it mean that all sites in Figure 1 use the same single snow profile as the prior? How did you determine the variance of the prior distribution?

That is correct. Since the SVS-2 are point scale simulations centred at SM02, all sites have the same prior. We apologize for neglecting key information in section 2.3, where a 120 member ensemble of SVS-2 outputs were generated for Woolley et al. (2024), where each member had a different parameterization for wind and basal vegetation effects, and thermal conductivity. The priors are then calculate using the mean and standard deviation of all ensemble members.

*A total of 120 different simulations were conducted with different combinations of wind and surface vegetation effects, and thermal conductivity parameterizations. These ensembles were used to generate the priors for the MCMC retrieval algorithm (see Section 3.5.1).*

6. Line 233: What does "top 30 ensemble members" mean? Are they the first 30 members closest to the study area, or those most similar to the measured snow profile?

Similarly to previous comment, some more information on the top 30 members of the ensemble was added to section 2.3:
*Test were also conducted in this study with the 30 ensemble members that had the best continuous ranked probability score (CRPS, see Woolley et al., 2024).*

7. Line 253: Could you use equations to describe the idea of DEMCZ for guiding the direction of chain evolution?

The methodology is a bit more complex than some equations. It is recommended to read section 2.2 of ter Braak and Vrugt (2008) to get all the details of the DEMCZ sampler. In this section, only the reasoning as to why it was selected over the more traditional DEMC or more efficient NUTS sampler is given. Users can also refer to the PyMC library: https://www.pymc.io/projects/docs/en/v5.6.1/api/generated/pymc.DEMetropolisZ.html#pymc.

8. Lines 266-270: The methodology for implementing the constraints can be mentioned here.

Clarification on the constraint method was added:
*Hard constraints were put on density, SSA and thickness between the layers. If those constraints were not met, the sampled values for these three parameters were rejected.*

9. Lines 335-337: Using the Arctic version of SVS-2 simulations, the uncertainties of MCMC retrieval results are reduced when the prior s.t.d. is reduced by narrowing down from all ensembles to the top 30 ensembles. This is reasonable.

Yes, this test was to determine the impact of prior standard deviation on the retrieved SWE uncertainty. This confirms that having better land surface models improves this retrieval method but has some costs if the land surface model is heavily biased with not much variability between the ensemble members, such as what is seen with SSA.

Additionally, in Table 1, is the s.t.d. of Hsnow(R) from the top 30 members (Arctic version) 9.5 or 0.95?

Thank you for bringing this up. The value is 9.5, all other values were reviewed and the error was with the STD of the 120 members. Values for SWE were also added in table 1 as proposed by reviewer Dr. Durand.

Actually, when comparing Fig.8(b) and Fig.7(b), I did not observe a reduction in the uncertainty (i.e., the range of the error bars on the Y-axis). Could you check the values?

The values have been checked. There was not a significant change in STD between the 120 and top 30 ensembles for the Arctic version. Table 2 shows the numerical values of that uncertainty and we see only an improvement of 1 mm between the two ensembles in the retrieved SWE uncertainty. This small difference is hardly visible between Figures 9b and 8b.

10. For the low correlation of the retrieved SWE to the measured SWE in Figs. 7 and 8, could you check the correlation between SWE (or SD) and the original radar observation inputted to MCMC? Are they highly-correlated or scattered?

The correlation for the retrieved SWE of Figure 9a was calculated between SWE and backscatter. As seen in Figure below, the points are scattered and $R^2$ is highly negative. This shows the complex nature of SWE retrieval where many combinations of snow properties can result in the same backscatter coefficient, thus needing proper constraints on the MCMC priors to retrieved realistic snow profiles.

11. The corresponding content of (a) and (b) is not labeled in Figure 9.

The legend now reads:
*Comparison of the retrieved SWE using the MCMC approach with priors coming from a) the top 30 ensemble members of the Arctic versions of SVS-2 (Woolley et al., 2024) and b) using the same top 30 ensemble members of the Arctic SVS-2, tripled prior uncertainty on the SSA ($\sigma_{SSA}$) and three additional radar backscatter ($\sigma^0$) observations (different incidence angles), emulating the number of observations the Terrestrial Snow Mass Mission (TSMM) would acquire. The error bars show the 1st and 3rd quartiles of the measured (x-axis) and posterior (y-axis) distributions.*

12. Lines 341-346: This result indicates the considerable impact of the grain size prior on SWE retrieval—not due to the accuracy of the mean, but rather due to the tolerance that allows the MCMC retrieval system to better match the observations. Increasing the variance of grain size indirectly enhances the influence of radar observations on the retrieval.

That is correct. This point is further discussed in the discussion section. In the results section, we only mention that the retrieved SWE is closer to the observed SWE.

[Figure]

13. Did Figure 10 utilize a single-angle radar backscatter, as in Figures 8 and 9?

*This is now explicitly mentioned in the figure legend. Figure 11a is with a single observation (same as Figure 9a), Figure 11b is with the higher SSA uncertainty and the four observations (same as Figure 10b), and the last one is the same configuration as Figure 9a without any constraints on the sampled snow properties to show that without them, it is possible to retrieve a SWE value closer to the measurement but the retrieved snow properties profile is not representative of what is truly measured in the field. Legend now reads:*

*Example from the SM site of prior distributions coming from the default (column a, same prior parameterization as Figure 9a) and Arctic (column b, same prior configuration as Figure 10b) versions of SVS-2 and retrieved posterior distributions from the MCMC method, for the different snow variables compared to the surveyed snow measurements. Column c) consists in the MCMC optimization using the default version of SVS-2, where no vertical constraints on snow properties were applied. $\sigma$ is the free parameter corresponding to the uncertainty of the radar backscatter measurement and was not measured in the field.*

14. Line 405: "other variables like thickness" -> Actually, I think what you really meant might be strategraphy, or strategraphy of layer thicknesses.

What was meant is layer thicknesses. This is also confirmed by the new experiment that was added, as suggested by the second reviewer. MCMC can easily change the thickness of each layer in order to fit the modelled backscatter to the observations. This confirms the need for good prior estimates and proper layering constraints.

*These results confirm that synthetic aperture radar (SAR) $\sigma^0$ is sensitive to SWE in the Ku-Band range, since, even with a poorly estimated microstructure (Figures 11 a and c), which is an important parameter that drives snow volume scattering in*
*that frequency range (Montpetit et al., 2024; Picard et al., 2022; King et al., 2018), other variables like thickness are tuned to fit the measured $\sigma^0$ (Figure 10 c), and can still achieve a reasonable SWE estimate compared to measurements. It should be noted that when SWE is poorly estimated by the prior, the posterior SWE estimate has a higher error (Figure 9), where SWE estimates are concentrated around the initial modelled SWE and do not diverge from that initial estimate. Also, further tests were done (not shown), where the uncertainty on the SWE was increase, by increasing uncertainty on thickness and density*
*individually and separately. Every tests resulted in underestimation of SWE, most likely due to the underestimation of SSA in the priors, which boosted the volume scattering of both layers. The most sensitive parameter in the MCMC model being thickness, it reduced the snow thickness to reduce the volume scattering and fit the modelled $\sigma^0$ with the measured $\sigma^0$, resulting in an underestimation of SWE.*

15. Lines 406-408: "It should be noted that when SWE is poorly estimated by the prior, the posterior SWE estimate has a higher
error (Figure 8), where SWE estimates are concentrated around the initial modeled SWE and do not diverge from that initial." -> The accuracy of the prior SWE does influence the SWE retrieval, and this is truly reflected in the likelihood calculation. However, for the case in Figure 10(a), I think the key point is that the default SVS-2 gives a highly-underestimated bottom-layer SSA (i.e., overestimated grain size); with a small variance, the system is forced to trust this value excessively. This resulted in the underestimation of SWE. Additionally, the default SWE prior is underestimated and has a low variance. This helped to
make things worse, slightly.

    The key point is not to overtrust the land surface model, allowing remote sensing to correct it. Trusting a wrong prior too much is the last thing to do, especially for radar-sensitive parameters.

Excellent point. This was added in the discussion:
*This is why it is important to have some knowledge of stratigraphic snow properties, e.g. number of snow layers, density and*
*SSA gradients, to constrain the MCMC method to valid snow properties without overfitting on the most sensitive parameters, and to not over-trust the initial prior estimates, i.e. not be too restrictive on the prior uncertainties.*

16. Lines 428-430: "Similarly, when comparing the outputs from both SVS-2 versions, the prior density estimates for the R layer of the default version (Figure 10a), do not allow to sample values close to the measured $\rho$snow, which prevents the MCMC method to properly sample other variables, such as SSA for the same layer." -> I do not fully agree that the snow
density influences SSA; rather, I think SSA influences itself. Or, they both influence both. This is because, in general, the sensitivity of radar signal to snow density is low.

More discussion on this topic was added with regards to your comment and comments made by reviewer 2. Here, the grain size parameter used by Improved Born Approximation (IBA) is correlation length which is given by (see Mätzler, 2002):

$$p_{\mathrm{c}} = \frac{4(1 - \frac{\rho_{\mathrm{snow}}}{\rho_{\mathrm{ice}}})}{\rho_{\mathrm{ice}}\, SSA} \tag{1}$$

So the scattering under IBA is driven by both SSA and density of snow. That said, the reviewer is right, the density does have a lesser impact but they do tend to influence each other in minimizing the simulated and observed backscatter.

*Similarly, when comparing the outputs from both SVS-2 versions, the prior density estimates for the R layer of the default version (Figure 11 a), do not allow to sample values close to the measured snow density ($\rho_{\mathrm{snow}}$), due to the lower prior*

*uncertainty, which prevents the MCMC method to properly sample other variables, such as SSA for the same layer, since volume scattering in the IBA model depends on both SSA and density.*

17. Line 500: It also indicates that remote sensing and land surface model can work together to mutually improve their accuracies. Ku-band radar is sensitive to snow depth and the SSA of the depth hoar layer, which can provide important information in regions with sparse measurements.

Absolutely, this is where the Terrestrial Snow Mass Mission aims at combining the radar measurements with the land surface model with a data assimilation scheme to keep improving both the model and the retrieved SWE from radar. This was added in the conclusion:

*This work also indicates that land surface models like SVS-2 and radar measurements can work together to mutually improve their accuracies. This is part of the TSMM concept where SVS-2 and the radar measurements will work together with a data*

*assimilation scheme to mutually improve their estimates, particularly in remote regions with little observations (Derksen et al., 2019).*

**1.3 Minor Comments:**

1. In the abstract, uncertainty should be described more clearly to distinguish it from the RMSE compared with in-situ data. For example, use the MCMC-estimated retrieval uncertainty.

The text in the abstract as been changed to clarify the "uncertainty" in the retrieved SWE:

*[...] we can retrieve SWE with an RMSE of 15.8 mm (16.4 %) and a MCMC-retrieved SWE uncertainty of 23.4 mm (25.2 %). [...]*

2. In the caption of Figure 3, second line: but->by.

Changed

3. Line 489: We also show -> We would also expect?

Correct, the sentence now reads:

*Given the results shown in this study, we should also expect that by allowing the priors to evolve in time and space, given the dynamic seasonal evolution of weather and snow conditions, a reduced number of iterations will be needed for the MCMC method to converge to a solution, thus improving computation efficiency.*

4. Line 503: "that influence the most the radar sigma0"—maybe change to "that influence the radar sigma0 the most"?

Changed

**References**

Derksen, C., Lemmetyinen, J., King, J., Belair, S., Garnaud, C., Lapointe, M., Crevier, Y., Burbidge, G., and Siqueira, P.: A Dual-Frequency Ku-Band Radar Mission Concept for Seasonal Snow, in: International Geoscience and Remote Sensing Symposium (IGARSS), pp. 5742–5744, https://doi.org/10.1109/IGARSS.2019.8898030, 2019.

Grünberg, I. and Boike, J.: Vegetation Map of Trail Valley Creek, Northwest Territories, Canada, https://doi.org/10.1594/PANGAEA.904270, 2019.

King, J., Derksen, C., Toose, P., Langlois, A., Larsen, C., Lemmetyinen, J., Marsh, P., Montpetit, B., Roy, A., Rutter, N., and Sturm, M.: The Influence of Snow Microstructure on Dual-Frequency Radar Measurements in a Tundra Environment, Remote Sensing of Environment, 215, 242–254, https://doi.org/10.1016/j.rse.2018.05.028, 2018.

Mätzler, C.: Relation between Grain-Size and Correlation Length of Snow, Journal of Glaciology, 48, 461–466, https://doi.org/10.3189/172756502781831287, 2002.

Montpetit, B., King, J., Meloche, J., Derksen, C., Siqueira, P., Adam, J. M., Toose, P., Brady, M., Wendleder, A., Vionnet, V., and Leroux, N. R.: Retrieval of Snow and Soil Properties for Forward Radiative Transfer Modeling of Airborne Ku-band SAR to Estimate Snow Water Equivalent: The Trail Valley Creek 2018/19 Snow Experiment, The Cryosphere, 18, 3857–3874, https://doi.org/10.5194/tc-18-3857-2024, 2024.

Picard, G., Löwe, H., Domine, F., Arnaud, L., Larue, F., Favier, V., Le Meur, E., Lefebvre, E., Savarino, J., and Royer, A.: The Microwave Snow Grain Size: A New Concept to Predict Satellite Observations Over Snow-Covered Regions, AGU Advances, 3, https://doi.org/10.1029/2021AV000630, 2022.

Porter, C., Howat, I., Noh, M.-J., Husby, E., Khuvis, S., Danish, E., Tomko, K., Gardiner, J., Negrete, A., Yadav, B., Klassen, J., Kelleher, C., Cloutier, M., Bakker, J., Enos, J., Arnold, G., Bauer, G., and Morin, P.: ArcticDEM - Mosaics, Version 4.1, https://doi.org/10.7910/DVN/3VDC4W, 2023.

Siqueira, P., Adam, M., Kraatz, S., Lagoy, D., Tarres, M. C., Tsang, L., Zhu, J., Derksen, C., and King, J.: A Ku-Band Airborne InSAR for Snow Characterization at Trail Valley Creek, in: 2021 IEEE International Geoscience and Remote Sensing Symposium IGARSS, pp. 2146–2149, https://doi.org/10.1109/IGARSS47720.2021.9554888, 2021.

ter Braak, C. J. F. and Vrugt, J. A.: Differential Evolution Markov Chain with Snooker Updater and Fewer Chains, Statistics and Computing, 18, 435–446, https://doi.org/10.1007/s11222-008-9104-9, 2008.

Woolley, G. J., Rutter, N., Wake, L., Vionnet, V., Derksen, C., Essery, R., Marsh, P., Tutton, R., Walker, B., Lafaysse, M., and Pritchard, D.: Multi-Physics Ensemble Modelling of Arctic Tundra Snowpack Properties, The Cryosphere, 18, 5685–5711, https://doi.org/10.5194/tc-18-5685-2024, 2024.

---

## Author Comment (AC2)

**Authors Response to Reviewers**

Montpetit et al.

**Correspondence:** Benoit Montpetit (benoit.montpetit@ec.gc.ca)

**1 Response to Reviewer #2**

**1.1 General Comments**

The authors present a study applying a Bayesian retrieval algorithm to estimate snow water equivalent, applied using the SMRT radiative transfer model, and priors derived from the SVS-2 land surface model, with airborne radar observations from the University of Massachusetts radar instrument, and intensive field data from snowpits. The algorithm achieves retrievals with RMSE less than 30 mm across several different experiments; as 30 mm threshold required for the proposed Terrestrial Snow Mass Mission (TSMM), these results are highly significant. The results do some really intriguing work exploring the sensitivity of the retrievals to prior estimates and their uncertainty, which will be of great value to the community. The results present a very interesting contribution to the literature of using these type of retrievals, and should be published directly. Adding one more aspect of the analysis in discussion is, in my opinion, critical to really understanding these results, however.

Retrieving SWE from radar measurements is an under constrained problem. To solve this, Bayesian methods optimally weigh priors estimates of snow properties and the radar observations. Prior configuration is a condition for obtaining meaningful results, and it's not clear whether the prior is accurately configured here. Specifically, the prior uncertainty (implicitly) specified for SWE appears to be set to be set too low. Thus, I recommend redoing the analyses with higher SWE uncertainty (see Major Comment #1). This should fit nicely with what the authors have already done in exploring the impact of the SSA uncertainty.

All other aspects of the analysis are really nicely described; the manuscript does a nice job dealing with the issues of both SSA bias and setting the SSA uncertainty too low, linking these in with model simulations specifically adapted for Arctic environment: this is a critical aspect of solving the SWE retrieval problem, and presents a significant advance over other recent efforts. Adding some analysis with more realistic SWE uncertainty will help to provide the context needed to understand the results shown here.

The authors would like to thank Dr. Durand for the positive, constructive and thorough review. All of the major/minor comments have been address below. Along with comments from reviewer #1, they significantly improved the quality of the

manuscript. In particular, the data methods and discussions sections were significantly improved. We also took note of the confusion in many aspects of the manuscript. We clarified many points and did extra tests, not shown in the manuscript, which greatly improve the comprehension of the study and analysis.

**1.2 Major comments**

1. The configuration of the SWE uncertainty in the SVS-2 model simulation ensemble and its impact on the results needs to be further discussed. The model is run in a "point" simulation mode, and the resulting ensemble of 120 members (sometimes down-selected to 30 members) is used to configure the prior. This ensemble is from a previous study (Woolley et al. 2024). The ensemble-derived prior is then applied to retrieve SWE at multiple spatial locations, but as far as I can tell, the total SWE simulated in the model is far too precise an estimate when applied as a prior in these various locations. The prior uncertainty appears to be shown in the width of the red rectangles in Figures 7-9; so in Figure 7a, e.g., the prior uncertainty appears to be 10 mm or less; Fig 7b it appears to be 5 mm, on the order of 5-10%. The uncertainty for the other experiments is similar. While the study does not quantify the spatial variability of snowpack SWE, in situ SWE appears (in e.g. Fig 7a) to range from around 60 mm to around 125 mm. Since this SWE from the simulation is being applied as a prior on each location, the uncertainty in simulated SWE should really be closer to the snowpit SWE spatial variability, so 50 mm (or more if you allow for bias). So, using the ensemble as an estimate of the SWE at these snowpits represents an almost order of magnitude mis-match in the uncertainty (~5-10 mm vs ~50 mm). This mis-match can have significant implications on the validity of the retrieval - basically the retrieval is implicitly assuming the prior estimates of SWE are already known to a high degree of precision. The fact that the SWE priors are not explicit, but rather represented as layer thickness and density can be navigated by increasing the respective uncertainties to make the resulting SWE uncertainty closer to the snowpit SWE standard deviation. In summary, I think it's key to address this issue needs to be addressed head-on in the paper, and to include an additional experiment (or revising one of the other experiments) with this increase to SWE uncertainty.

We agree with the reviewer that this needs to be clarified. Since the priors used in the MCMC model are thickness and density for the two layers, what is represented in Figure 8 to 10 (Figure numbering changed due to additional Figure) is not the uncertainty in the prior used in the MCMC model but the variability in simulated bulk SWE by SVS-2. The authors kept the figures as is, but clarified this point since it shows that MCMC can generate much more variability than SVS-2 in bulk SWE, which is expected given the different spatial scales at which the model and radar measurements. The uncertainty on SWE "prior" is represented in Figure 11 where we have an uncertainty closer to 30 mm. Some of this was already discussed in Section 5 but we have done further tests where we increased the uncertainty on both thickness and density individually and at the same time. These tests are not shown in the manuscript to avoid adding confusion and too much information but additional information from those tests was added to the discussion. We have seen that the retrieval is very sensitive to thickness, where a small change in thickness can compensate for a big difference in density or SSA, thus fitting the observed backscatter but, in this case where SSA and density is underestimated, SWE is also underestimated by reducing thickness of both layers to

55  compensate for the high volume scattering from low SSA values.

*Results: Figure 6 shows the same distributions for snow water equivalent (SWE). We see that Canadian Soil Vegetation Snow version 2 (SVS-2) tends to overestimate SWE for the rounded grains (R) layer and overestimate SWE for the depth hoar (DH) layer. Variability in modelled layered SWE for both versions of SVS-2 is similar to what is observed in the field. This results in a very narrow range of modelled bulk SWE that fit very well with the observations. This tends to indicate that SVS-2 reproduces*

60  *the bulk SWE properly over Trail Valley Creek (TVC) but has more difficulty in properly representing proper SWE stratigraphy. Figure 6c also shows that the higher uncertainties and the over- and underestimations of the layered SWE tend to cancel out for the bulk properties. It should be noted that the distributions in Figure 6c do not realistically represent the priors and the uncertainty on SWE since SWE is not an explicit variable used in the Markov Chain Monte Carlo (MCMC) model.*

*Discussion: These results confirm that synthetic aperture radar (SAR) radar backscatter ($\sigma^0$) is sensitive to SWE in the Ku-Band*

65  *range, since, even with a poorly estimated microstructure (Figures 11 a and c), which is an important parameter that drives snow volume scattering in that frequency range (Montpetit et al., 2024; Picard et al., 2022; King et al., 2018), other variables like thickness are tuned to fit the measured $\sigma^0$ (Figure 11 c), and can still achieve a reasonable SWE estimate compared to measurements. It should be noted that when SWE is poorly estimated by the prior, the posterior SWE estimate has a higher error (Figure 9), where SWE estimates are concentrated around the initial modelled SWE and do not diverge from that initial*

70  *estimate. Also, further tests were done (not shown), where the uncertainty on the SWE was increased, by increasing uncertainty on thickness and density individually and separately. Every tests resulted in underestimation of SWE, most likely due to the underestimation of Specific Surface Area (SSA) in the priors, which boosted the volume scattering of both layers. The most sensitive parameter in the MCMC model being thickness, it reduced the snow thickness to reduce the volume scattering and fit the modelled $\sigma^0$ with the measured $\sigma^0$, resulting in an underestimation of SWE. [...] A way to potentially solve the issue of the*

75  *high sensitivity to layer thickness would be to use SWE as a prior directly and infer snow density and thickness from published relationships between SSA and density (Domine et al., 2007). This will be further tested in future experiments in the context of Terrestrial Snow Mass Mission (TSMM).*

2. It is really remarkable that there is so little variability among the SWE retrievals. Fig 7a shows that all the retrievals except one seem to fall in a range of 75±5 mm, when the snowpits range from 60 to 125 mm. At first glance, this appears to be

80  totally expected, and a result of having the SWE uncertainty set so low: if you start with a prior SWE that's known to within 5 %, you're not likely to change that very much, regardless of the measured backscatter value. However, Fig 7a also shows that the retrieval is adjusts fairly significantly, moving down from ∼90 mm to 75 mm. Other experiments show less change in the average retrieved SWE (the Arctic SVS-2, the top 30 ensemble members, the increased SSA uncertainty, and the multiple observation angles), but across all six experiments (Fig 7a, 7b, 8a, 8b, 9a, 9b) there is almost no variability across the retrieved

85  SWE values. Is this due to lack of variability in the observed backscatter? Does the retrieved soil properties from Montpetit et al. 2024 play a role? Please discuss some possible reasons for this lack of much variability among the retrievals in the manuscript.

It is difficult to speak of "SWE prior" when discussing this experiment since SWE is not an explicit variable in MCMC. That said, we acknowledge the confusion coming from the red shaded area of Figure 8 to 10, and this has been clarified. The "SWE prior" uncertainty is more of the order of 30 mm (30 %)First, we believe the lack of variability comes from the small sensitivity to density and the very high sensitivity to thickness. This results in MCMC adapting thickness and SSA which are the most sensitive parameters to the radar measurements. It is possible that the extra backscatter measurements at the different incidence angles do not introduce considerably more sensitivity to SWE, and this is why TSMM is planning an additional higher frequency Ku-Band measurement, which will be more sensitive to SWE and volume scattering. This is now further discussed in the discussion section with the additional tests that were conducted. Second, in Montpetit et al. (2024), it is shown that adapting the soil properties to different land cover types does improve error between simulated and measured backscatter, but it should not have a considerable impact on the SWE retrieval using MCMC since the uncertainty in the radar backscatter (previously $\sigma$, now $\delta$ in the revised manuscript) are equal or higher to the improvement of the error when considering a single set of soil properties. This has been added to the discussion.

*This result also shows that, even though the lower Ku-Band frequency (13.5 GHz) is sensitive to SWE and snow volume scattering, it still has a high sensitivity to surface scattering at the soil-snow interface, which could explain why there is not significant spread among the retrieved SWE. The lack of spread in the retrieved SWE can also be explained by the low uncertainty on the thickness priors of both layers, which are the most sensitive parameters in the MCMC model, but are restrained to a narrow range of thickness values. Also, the fact that volume scattering is less impacted by density, than it is by SSA, reduces the potential of sampling a wider range of SWE values, which is a function of snow thickness and density.*

*This result also shows that, even though the lower Ku-Band frequency (13.5 GHz) is sensitive to SWE and snow volume scattering, it still has a high sensitivity to surface scattering at the soil-snow interface, which could explain why there is not significant spread among the retrieved SWE. The lack of spread in the retrieved SWE can also be explained by the low uncertainty on the thickness priors of both layers, which are the most sensitive parameters in the MCMC model, but are restrained to a narrow range of thickness values. Also, the fact that volume scattering is less impacted by density, than it is by SSA, reduces the potential of sampling a wider range of SWE values, which is a function of snow thickness and density. The higher error coming from using optimized effective soil properties (increased RMSE of 0.4 dB) for all the TVC domain instead of site specific variables (see Fig.13 of Montpetit et al., 2024), propagates in the uncertainty of the $\sigma^0$ measurement and impacts the retrieved snow properties. Since the uncertainty on the modelled $\sigma^0$ values ($\delta$) are of the order of 1.5 dB, compared to a change in RMSE of 0.4 dB, it is unlikely that considering site specific soil properties will have a significant impact on the retrieved SWE in this study. That said, it was shown by (Montpetit et al., 2024) that soil cannot be neglect for SWE retrievals at Ku-band, and that its properties must be properly estimated.*

**1.3 Minor comments**

1. Line 64: Please add that for these Bayesian methods, it's key to correctly specify the SWE uncertainty, which in prior studies was done by specifying layer thickness and density uncertainties.

*A sentence was added to mention this:*

*It is key for these methods to correctly specify SWE uncertainty, where it was achieved by specifying layer and density uncertainties.*

2.Line 92, Figure 1: Please comment on the difference in backscatter between the two images. Presumably these are made at different incident angles? At first blush, it looks as if nearly the same scene produced significantly different backscatter, so may need some additional discussion.

More details is now given in the text with regards to the SAR mosaic in the background. As the reviewer mentions, the image was done by overlapping two different airborne passes in the same direction, i.e. the far range of the first image meets the near range of the second :

*The radar image in Figure 1 consists in a mosaic of two different airborne passes, flown in the same direction, acquired by the University of Massachusetts (UMASS) radar system (Section 2.1), where the near range acquisitions (higher backscatter) of the first pass, done at steeper incidence angles, meets the far range acquisitions (lower backscatter) of the second pass, made at shallower incidence angles.*

3.Line 92, Figure 1: the combination of using a point-scale SVS-2 simulation and the reference to the Canadian High Resolution Deterministic Prediction System were a little confusing. I thought at first that CHRDPS was used in the modeling setup. I recommend just removing that grid and mention of CHRDPS if that data is not used in the study

This has been removed. Along with the comments from reviewer #1, the figure has been updated and the legend now reads:
*Figure 1. Sites sampled during the January campaign of the TVC 2018/19 experiment. Squares correspond to a 100 m x 100 m around the central surveyed snowpit (see Section 2.2). Background images are two overlapped UMASS Ku-Band radar images corresponding to two different flight passes acquired November 14, 2018 (left, Siqueira et al., 2021), the 2 m ArcticDEM (center, Porter et al., 2023), and the vegetation classification (right, Grünberg and Boike, 2019).*

4.Clarify what "top 30" means in methods, the first time it's introduced (line 233) not in results (line 330). Is this derived to be different for each snowpit, or is it the same 30 for all snowpits? Is it based only on SWE, or on SWE and SSA or other properties? Is it e.g. the 30 SWE values closest to the true average snowpit SWE, e.g.?

Yes, some key elements of the SVS-2 ensembles were missing from Section 2.3. To clarify what the "top 30" means, this was added in Section 2.3:
*Test were also conducted in this study with the 30 ensemble members that had the best continuous ranked probability score (CRPS, see Woolley et al., 2024)*

5.Line 226: Please note that when deriving SWE priors from the ensemble, the resulting prior SWE uncertainty is important, and cannot be directly obtained from the model ensemble estimates of layer thickness and density uncertainty, due to possible ensemble correlations of these quantities.

This is absolutely true. The re shaded areas shown in Figures 7 to 9 represent the variability in modelled bulk SWE by the different SVS-2 version. This variability does not translate into the uncertainty that was given in the MCMC model. Figure 6 shows how bulk SWE is not that variable in its outputs but the layered SWE is. When looking at Figure 11, it is clear that the prior uncertainty on SWE is much larger than the modelled SWE variability. The authors preferred to keep the modelled variability in Figures 8 to 10 since it shows that MCMC can reproduce a lot more variability than SVS-2, which is expected. Nonetheless, this has been clarified in the figure legend and additional text was added in the discussion.

*The red shaded areas show the 1st and 3rd quartiles of the SVS-2 distributions (Figure 6c).*

6.Line 237, Table 1: Please add the SWE ensemble standard deviation and mean values here.

Values were added to Table 1.

7.Line 245: Define all symbols in equation 1. Should also note the confusing aspect of $\sigma$ being used to represent both uncertainty, and backscatter.

The symbols are now all defined and $\sigma$ for uncertainty was modified for $\delta$ throughout the text.

*[...]*

$$l(\sigma^0_{mes}, \delta | \sigma^0_{sim}) = -\frac{1}{2}\left(\frac{\sigma^0_{mes} - \sigma^0_{sim}}{\sigma}\right)^2 - \ln(\sqrt{2\pi}) - \ln(\delta) \tag{1}$$

*where $l(\sigma^0_{mes}, \delta | \sigma^0_{sim})$ is the likelihood metric between the measured $\sigma^0$ ($\sigma^0_{mes}$) and simulated $\sigma^0$ ($\sigma^0_{sim}$), given an uncertainty on the measured $\sigma^0$ ($\delta$).*

8.Line 278, Figures 3, 4 and 5. Calling the red line the "posterior" is sure to create confusion, as "posterior" is usually meant to be an output of the retrieval, and this is a normal distribution with mean and standard deviation derived from the snowpits

This was changed to "Snowpits" as in Figure 11.

9.Line 280, section 4.1: it is critical to provide the basic statistical summaries of the snowpit SWE. How variable was the SWE? This must be quantified, given the study objectives.

An extra figure was made for SWE (see Figure 6) with additional text to describe the SWE variability.
*Figure 6 shows the same distributions for SWE. We see that SVS-2 tends to overestimate SWE for the R layer and overestimate SWE for the DH layer. Variability in modelled layered SWE for both versions of SVS-2 is similar to what is observed in the field. This results in a very narrow range of modelled bulk SWE that fit very well with the observations. This tends to indicate that SVS-2 reproduces the bulk SWE properly over TVC but has more difficulty in properly representing proper SWE stratigraphy.*

*Figure 6c also shows that the higher uncertainties and the over- and underestimations of the layered SWE tend to cancel out for the bulk properties. It should be noted that the distributions in Figure 6c do not realistically represent the priors and the uncertainty on SWE since SWE is not an explicit variable used in the MCMC model.*

10.Line 280, section 4.1: it is critical to provide the statistical summary of the simulated SWE. This can't be easily retrieved from the depth and density histograms, because they may be cross-correlated. The little red boxes in Figure 7-9 do not provide enough quantitative information, given that the paper is aimed at SWE accuracy.

*Yes, similarly to previous comment, more description on SWE variability was added in Section 4.1.*
*Figure 6 shows the same distributions for SWE. We see that SVS-2 tends to overestimate SWE for the R layer and overestimate SWE for the DH layer. Variability in modelled layered SWE for both versions of SVS-2 is similar to what is observed in the field. This results in a very narrow range of modelled bulk SWE that fit very well with the observations. This tends to indicate that SVS-2 reproduces the bulk SWE properly over TVC but has more difficulty in properly representing proper SWE stratigraphy. Figure 6c also shows that the higher uncertainties and the over- and underestimations of the layered SWE tend to cancel out for the bulk properties. It should be noted that the distributions in Figure 6c do not realistically represent the priors and the uncertainty on SWE since SWE is not an explicit variable used in the MCMC model.*

11.Line 280, section 4.1: From comparing the various figures, it looks as if the simulation underestimates total depth, and overestimates total density, and ends up with a slight bias in SWE. Please comment on this and clarify these biases

*SVS-2 tends to underestimate density for both layers for a total underestimation of bulk density. SVS-2 overestimates thickness of the rounded grain layer (R) but underestimates the depth hoar thickness. SVS-2 slightly overestimates the total thickness. This results in a very good estimations of bulk SWE but a closer look shows an underestimation of the SWE from the DH layer and an overestimation of the SWE for the R layer.*
*We see that SVS-2 tends to overestimate SWE for the R layer and overestimate SWE for the DH layer. Variability in modelled layered SWE for both versions of SVS-2 is similar to what is observed in the field. This results in a very narrow range of modelled bulk SWE that fit very well with the observations. This tends to indicate that SVS-2 reproduces the bulk SWE properly over TVC but has more difficulty in properly representing proper SWE stratigraphy.*

12.Line 280, section 4.1: Please directly compare the measured snowpit SWE and the ensemble SWE, and discuss implications of SWE uncertainty estimation on the retrieved SWE.

*New figure added for layered and bulk SWE, with description in Section 4.1 and added discussion in Section 5.2.*
*Figure 6 shows the same distributions for SWE. We see that SVS-2 tends to overestimate SWE for the R layer and overestimate SWE for the DH layer. Variability in modelled layered SWE for both versions of SVS-2 is similar to what is observed in the field. This results in a very narrow range of modelled bulk SWE that fit very well with the observations. This tends to indicate that SVS-2 reproduces the bulk SWE properly over TVC but has more difficulty in properly representing proper SWE stratigraphy.*

*Figure 6c also shows that the higher uncertainties and the over- and underestimations of the layered SWE tend to cancel out for the bulk properties. It should be noted that the distributions in Figure 6c do not realistically represent the priors and the uncertainty on SWE since SWE is not an explicit variable used in the MCMC model.*

215    13.Line 315, Figure 7: To me, it looks like the basic result that the high bias in SSA leads to an underestimation of scattering which then leads the retrievals to bias the SWE low (Fig 7a). However, this bias does not persist in all other scenarios; the Arctic adapted SVS-2 has less bias in SSA and leads to less bias in SWE. It would be nice to comment further on this.

That is correct. The bias in SVS-2 SSA is discussed in Section 5.1 as a current limitation of SVS-2, but additional text was added with regards to bias in SSA (underestimation) leading to an overestimation of volume scattering, which results in an
220    underestimation of SWE since MCMC reduces the thickness of each layer to account for too much scattering from the grain size.

*Also, further tests were done (not shown), where the uncertainty on the SWE was increased, by increasing uncertainty on thickness and density individually and separately. Every tests resulted in underestimation of SWE, most likely due to the underestimation of SSA in the priors, which boosted the volume scattering of both layers. The most sensitive parameter in the*
225    *MCMC model being thickness, it reduced the snow thickness to reduce the volume scattering and fit the modelled $\sigma^0$ with the measured $\sigma^0$, resulting in an underestimation of SWE.*

14.Line 315, Figure 7, 8, 9: the error bars on the measurements are confusing. In this context, error bars like this imply an uncertainty in the measured value at the snowpit. However, if I understand, these are taken as the spatial variability of the measured values, which I don't think is the same as the uncertainty. The uncertainty is how accurately you think you measured
230    SWE at each place, whereas spatial variability is how much SWE varies from place to place. Please derive a reasonable SWE uncertainty and change the error bars

It is true that the error bars are related to spatial variability of SWE. In this context, the spatial variability within a 100 m footprint of the radar measurements, for a given site, is considered part of the snow measurements uncertainty. This is why those error bars are there. Other metrics of uncertainty could be considered for these error bars like measurement uncertainty
235    and vertical variability within the different Snow Micropenetrometer profiles taken but the spatial variability is considered here to have the highest impact on measured radar backscatter. Also, since the spatial variability within the radar footprint is not always best represented by a normal distribution, the error bars represent the quartile deviation (Q1 and Q3).

15.Line 315, Figure 7, 8 9: the error bars on the model seems to have been drawn wrong on figure a? The dashed line appears
240    at the bottom of the shaded area, but I think the dashed line should be in the middle of the shaded box?

The shaded boxes were modified in Figures 8 to 10 to better represent the SVS-2 SWE uncertainty used in the MCMC model. That said, the shaded box represent the quartile deviation (Q1 and Q3). Since the SVS-2 SWE distribution is skewed, the

*dashed line is not centred in the shaded area.*

245   16.Line 320: It's unclear what SWE uncertainty means in this context. Is this derived from the Markov Chains? It almost reads as if this quantity is being calculated across the retrievals at the various snowpits, but that is spatial variability rather than uncertainty, which is not the same thing. If uncertainty were calculated at each snowpit, then you'd have a different uncertainty for each snowpit, as you ought, and so the Table 2 could summarize the average uncertainty, which may be what is there; but in any case, please clarify!

250   The uncertainty metric used for the retrieved and measured SWE is the quartile deviation rather than the usual standard deviation. This is due to the fact that the posterior distributions from MCMC are not all normal distributions and are often skewed. The quartile deviation is better suited to estimate the retrieved uncertainty. Values in Table 2 consist in the mean uncertainty over all sites. This has been clarified in the text and table legend.
*Uncertainty is defined as the quartile deviation, and the values consist in the mean value over all sites.*

255   17.Line 350, comparing Figure 8 vs Figure 9: why don't the error bars look the same here? In Figure 8a, the default SVS-2 looks much wider than Figure 8b. But as I read the paper, this shouldn't change when we go to Figure 9, correct? But Figure 9 looks much smaller?

The SVS-2 uncertainty is represented by the red shaded area. Figure 9b, 10a and 10b all have the same prior, the Top 30 Arctic version, where as Figure 9a uses the Top 30 Default version. Thus the shaded area is different between Figure 9a and all other
260   figures, which have the same shaded areas.

18.Line 360: I found Figure 10 a bit confusing. So this is at a single pit? In that case, the snowpits pdf is not a reasonable thing to show, right? Why not just show the observation at that particular pit? Same comment in Figure 11, why not show the measured value rather than the median over all pits? Those are not very relevant except this one. Similarly, the text refers to
265   bias which is usually an average over many error samples. But in this case, you should just have one error, which is equal to the estimate of each quantity from the MCMC and the measured value.

This has been clarified in Section 2.2. Was is shown is the distribution of measurements for a given site which consists in a 100 m x 100 m footprint ( 20 profiles total between SMP and reference snowpit).
*All profiles (snowpits, SMP and MagnaProbe profiles) per site are then used to generate a statistical representation of snow*
270   *conditions within the radar footprint, with a measured snow uncertainty represented by the spatial variability within the footprint. Spatial variability consists in the largest uncertainty within the footprint compared to snow parameter measurement uncertainty, the latter can thus be neglected.*

19.Line 373: What do min and max values mean for each iteration in Figure 11?

The min/max values for each iteration is the min/max values sampled across the seven parallel chains. This has been clarified

275   in Figure legend:

*Full horizontal lines consist in median values from measurements (snowpits) and the dashed horizontal lines consist in the mean of the priors. The min/max values consist in the minimum and maximum properties sampled at each iteration between the seven parallel sampling chains.*

20. Line 373: Please show the prior values in Figure 11.

280   Prior values were added to the figure as a dashed line.

**References**

Domine, F., Taillandier, A.-S., and Simpson, W.: A Parameterization of the Specific Surface Area of Seasonal Snow for Field Use and for Models of Snowpack Evolution, Journal of Geophysical Research: Earth Surface, 112, https://doi.org/10.1029/2006JF000512, 2007.

Grünberg, I. and Boike, J.: Vegetation Map of Trail Valley Creek, Northwest Territories, Canada, https://doi.org/10.1594/PANGAEA.904270, 2019.

King, J., Derksen, C., Toose, P., Langlois, A., Larsen, C., Lemmetyinen, J., Marsh, P., Montpetit, B., Roy, A., Rutter, N., and Sturm, M.: The Influence of Snow Microstructure on Dual-Frequency Radar Measurements in a Tundra Environment, Remote Sensing of Environment, 215, 242–254, https://doi.org/10.1016/j.rse.2018.05.028, 2018.

Montpetit, B., King, J., Meloche, J., Derksen, C., Siqueira, P., Adam, J. M., Toose, P., Brady, M., Wendleder, A., Vionnet, V., and Leroux, N. R.: Retrieval of Snow and Soil Properties for Forward Radiative Transfer Modeling of Airborne Ku-band SAR to Estimate Snow Water Equivalent: The Trail Valley Creek 2018/19 Snow Experiment, The Cryosphere, 18, 3857–3874, https://doi.org/10.5194/tc-18-3857-2024, 2024.

Picard, G., Löwe, H., Domine, F., Arnaud, L., Larue, F., Favier, V., Le Meur, E., Lefebvre, E., Savarino, J., and Royer, A.: The Microwave Snow Grain Size: A New Concept to Predict Satellite Observations Over Snow-Covered Regions, AGU Advances, 3, https://doi.org/10.1029/2021AV000630, 2022.

Porter, C., Howat, I., Noh, M.-J., Husby, E., Khuvis, S., Danish, E., Tomko, K., Gardiner, J., Negrete, A., Yadav, B., Klassen, J., Kelleher, C., Cloutier, M., Bakker, J., Enos, J., Arnold, G., Bauer, G., and Morin, P.: ArcticDEM - Mosaics, Version 4.1, https://doi.org/10.7910/DVN/3VDC4W, 2023.

Siqueira, P., Adam, M., Kraatz, S., Lagoy, D., Tarres, M. C., Tsang, L., Zhu, J., Derksen, C., and King, J.: A Ku-Band Airborne InSAR for Snow Characterization at Trail Valley Creek, in: 2021 IEEE International Geoscience and Remote Sensing Symposium IGARSS, pp. 2146–2149, https://doi.org/10.1109/IGARSS47720.2021.9554888, 2021.

Woolley, G. J., Rutter, N., Wake, L., Vionnet, V., Derksen, C., Essery, R., Marsh, P., Tutton, R., Walker, B., Lafaysse, M., and Pritchard, D.: Multi-Physics Ensemble Modelling of Arctic Tundra Snowpack Properties, The Cryosphere, 18, 5685–5711, https://doi.org/10.5194/tc-18-5685-2024, 2024.

---

## Author Response (AR2)

**Authors Response to Editor**

Montpetit et al.

**Correspondence:** Benoit Montpetit (benoit.montpetit@ec.gc.ca)

1. The Short summary system section includes abbreviations. Please adapt your short summary avoiding abbreviations to make it better understandable for non-experts and please pay attention to use only 500 characters including spaces.

Abbreviations and character count have been addressed.

- 2. The ROR database lists the institution of the corresponding author but with a different city than given in the manuscript.
- Please clarify whether the ROR Environment and Climate Change Canada (Ottawa, Canada) is still correct.

In the manuscript, only the province (Ontario) of the corresponding author was given since he has moved to a different section of ECCC which is in Toronto (Ontario) and no longer in Ottawa (Ontario).

2. Some figures are misplaced. Please place all the figures either in a text or after the references.

Figures are well positioned in the .tex file and the width of the figures were modified to ensure they are well positioned in the pdf. If this is an issue, I can quickly change this so that the figures are after the references in the pdf.

3. Figure 1 contains maps. To clarify whether a copyright statement or a credit must be given in the map itself or in the caption, we differentiate between (a) maps entirely created by you, (b) maps created by you but based on layers reused from other originators, or (c) maps simply reused from other originators. An example for (a) is a digital elevation model (DEM) purely based on measurement points collected by you and derived by using a software product. If you use an existing map layer from another originator as a basis for significantly enriching the map with your own content, this would be an example for case (b). Case (c) could be a pure reproduction of Google Maps where your own contribution is rather small (e.g. a city map where you only added a few marks for your study locations). If the map was entirely created by you (case a), there is no need to change the caption or map. Please simply inform us. To the contrary, if your map follows cases (b) or (c), please let us know whether the map is distributed under public domain. If yes, please do not include a copyright statement (copyright is waived) but consider adding a credit to the map or caption. However, if your map follows cases (b) or (c) and is not distributed under public domain, please include at least a credit or even a copyright statement (e.g. © Google Maps), if this is required by the map provider, in the map itself or in the caption.

Figure 1 corresponds to scenario a) where public facing data was used by the main author to generate the map shown in Figure 1. Reference to the datasets used in the Figure is included in the caption.